# A haplotype-based evolutionary history of barley domestication

Yu Guo[1], Murukarthick Jayakodi[1], Axel Himmelbach[1], Erez Ben-Yosef[2], Uri Davidovich[3], Michal David[4], Anat Hartmann-Shenkman[4], Mordechai Kislev[5], Tzion Fahima[6], Verena J. Schuenemann[7,8], Ella Reiter[9], Johannes Krause[10], Brian J. Steffenson[11], Nils Stein[1,12], Ehud Weiss[4]✉ & Martin Mascher[1,13]✉

Barley is one of the oldest cultivated crops, with a complex evolutionary and domestication history[1]. Previous studies have rejected the idea of a single origin and instead support a model of mosaic genomic ancestry[2,3]. With increasingly comprehensive genome data, we now ask where the haplotypes — the building blocks of this mosaic — originate, and whether all domesticated barleys share the same wild progenitors or whether certain wild populations contribute more heavily to specific lineages. To address these questions, we apply a haplotype-based approach to investigate the genetic diversity and population structure of wild and domesticated barley. We analyse whole-genome sequences from 682 genebank accessions and 23 archaeological specimens, tracing the spatiotemporal origins of haplotypes and identifying wild contributors during domestication and later gene flow events. Ancient DNA supports our genome-wide findings from modern samples. Our results suggest that a founding domesticated population emerged in the Fertile Crescent during a prolonged period of pre-domestication cultivation. A key practical insight is that the high haplotype differentiation among barley populations — arising independently, or layered on top, of selection — poses challenges for mapping adaptive loci.

Barley (*Hordeum vulgare*) is an old crop. It is mentioned in some of the earliest records of human writing (3100 BCE)[4]. By that time, plant cultivation was older than written language is now. Much of what we know about the early stages of the domestication and dispersal of barley and other crops comes from archaeological specimens, the earliest dated to 10,000 years before present (BP)[1,5]. These are mainly charred grains from which archaeobotanists can infer hallmarks of domestication such as loss of spike brittleness[1]. Molecular genetics has complemented these findings by identifying domestication genes and tracing the origins of their alleles in wild populations[6]. With the advent of affordable whole-genome sequencing, our ability to study crop evolution at high resolution has greatly improved[7]. New methods, such as the pairwise sequentially Markovian coalescent (PSMC), allow researchers to infer historical population dynamics from extant genome[8]. More recently, tools such as IntroBlocker have been developed to define ancestral haplotype groups (AHGs), enabling inference of local ancestry at the haplotype level rather than at the whole-genome scales[9]. These advances make it possible to ask not only where barley was domesticated but also how different genomic regions in domesticated barley trace back to their wild ancestors. Finally, ancient DNA sequences provide valuable insights into past genetic diversity, although their use is limited by the often poor preservation of plant material in many climates[10].

For decades, researchers have sought to identify single centres of domestication using molecular markers, as in the case of einkorn wheat[11]. However, such a model has been increasingly challenged[12]. In barley, strong evidence refutes a monophyletic origin. For example, two independent mutations causing loss of spike brittleness — an essential domestication trait — are associated with geographically distinct wild progenitors[6]. Genome-wide data further support a mosaic ancestry model, in which domesticated barley derives from multiple wild populations[2,13,14]. Earlier studies often relied on reduced-representation sequencing or markers ascertained in domesticated lines, limiting their resolution. With the availability of high-quality reference genomes[15] and broader sequencing of wild and ancient barleys[16,17], we can now revisit barley domestication with greater precision.

Here we used whole-genome sequences from 682 genebank accessions and 23 ancient specimens to reconstruct the haplotype-level ancestry of domesticated barley. By assigning local genomic regions to their closest wild relatives, we asked which parts of the domesticated genome derive from which wild ancestors; whether certain wild populations contribute disproportionately to domesticated

[1]Leibniz Institute of Plant Genetics and Crop Plant Research (IPK), Seeland, Germany. [2]Department of Archaeology and Ancient Near Eastern Cultures, Tel Aviv University, Tel Aviv, Israel. [3]Institute of Archaeology, The Hebrew University of Jerusalem, Jerusalem, Israel. [4]The Martin (Szusz) Department of Land of Israel Studies and Archaeology, Bar-Ilan University, Ramat-Gan, Israel. [5]Faculty of Life Science, Bar-Ilan University, Ramat-Gan, Israel. [6]Institute of Evolution, University of Haifa, Haifa, Israel. [7]Department of Environmental Sciences, University of Basel, Basel, Switzerland. [8]Institute of Evolutionary Medicine, University of Zurich, Zurich, Switzerland. [9]Archaeo- and Paleogenetics Group, University of Tübingen, Tübingen, Germany. [10]Department of Archaeogenetics, Max Planck Institute for Evolutionary Anthropology, Leipzig, Germany. [11]Department of Plant Pathology, University of Minnesota, St. Paul, MN, USA. [12]Institute of Agricultural and Nutritional Sciences, Martin Luther University Halle-Wittenberg, Halle, Germany. [13]German Centre for Integrative Biodiversity Research (iDiv) Halle-Jena-Leipzig, Leipzig, Germany. ✉e-mail: Ehud.Weiss@biu.ac.il; mascher@ipk-gatersleben.de

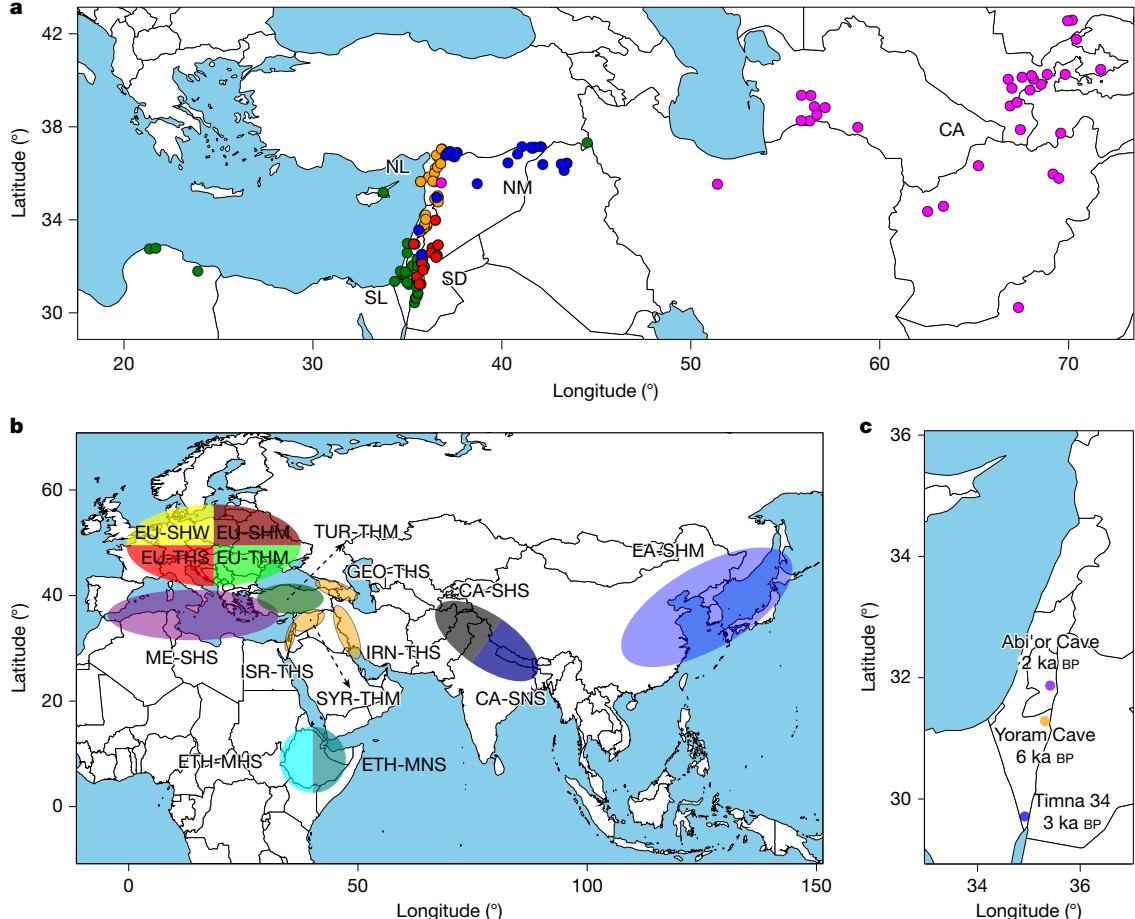

**Fig. 1 | Diversity panel of wild and domesticated barley. a**, Collection sites and population structure of 143 wild barley genotypes with precise geographical locations. The coloured dots show the results of model-based ancestry estimation with ADMIXTURE (the number of ancestral populations ($K$) = 5, predominant ancestry component) and are plotted at approximate collection sites. Jitter was added to avoid overlaps between nearby accessions. Only unadmixed samples, that is, those whose major ancestry components was 0.85 or more are shown. CA, Central Asia; NL, northern Levant; NM, northern Mesopotamia; SD, Syrian desert; SL, southern Levant. **b**, Assignment to macrogeographical regions of 15 populations inferred from GBS data of

19,778 domesticated barley genotypes[24]. The population names encode the most common origin of samples and their predominant morphological and phenological characters (row type, lemma adherence and annual growth habit) as detailed in Supplementary Table 6. **c**, Archaeological sites at which ancient barley grains used for ancient DNA extractions were found. Ages of the samples, as determined by radiocarbon dating, are indicated in the figure. Geographical outlines in panels **a**–**c** were obtained from the R package 'maps' (https://CRAN.R-project.org/package=maps), which uses public domain base map data (under a GNU General public license: version 2).

lineages; and how haplotypes were reshuffled through domestication and post-domestication gene flow. Our integrative, haplotype-based approach sheds new light on the origins and evolutionary assembly of one of the world's most important crops.

## Structure and divergence of wild barley

We started with the assumption that the present-day population structure of wild barley is related to what it was when human beings began to grow barley. Wild barley (*H. vulgare* subsp. *spontaneum*) is a genetically diverse taxon that occurs throughout western Asia. We analysed a total of 380 wild barley accessions, many of them from the Wild Barley Diversity Collection[17], which had been sequenced to tenfold coverage with Illumina short reads (Supplementary Tables 1 and 2). Previous studies on wild barley agree on the fact that isolation by distance is the main driver of population differentiation in wild barley[14,18]. Using model-based ancestry estimation complemented by principal component analysis (PCA), we divided our panel into five populations whose geographical distributions roughly trace a path from the southern Levant, via the Syrian Desert, the northern Levant, northern Mesopotamia and Central Asia (Fig. 1a, Extended Data Fig. 1a,b, Supplementary

Table 3 and Supplementary Fig. 1). These populations had different levels of diversity (Extended Data Fig. 1c and Supplementary Table 4). Low diversity in the Syrian Desert populations, which was accompanied by high differentiation from other populations, might be explained by higher genetic drift in the Syrian Desert (Extended Data Fig. 1d).

If there were no recombination and gene flow, the number of sequence variants between two genomes would inform directly about divergence times. Three examples illustrate that this simple model is not applicable in barley: when we counted single-nucleotide polymorphisms (SNPs) in 1-Mb windows and plotted the SNP distribution, we observed, between some pairs of samples, local differences in divergence times, most prominently between distal and proximal regions (Extended Data Fig. 2a,b). In barley and its relatives wheat and rye, proximal non-recombining regions, so-called genetic centromeres, are extensive, have fewer genes and drastically reduced recombination[19–21]. In domesticated barley, sequence diversity in these regions is lower too[14,19]. The situation in wild barley is more nuanced. Looking only at between-population comparisons, the distributions of divergence times were unimodal in distal regions of all chromosomes with a peak at around 600 thousand years before present (ka BP; Fig. 2a and Extended Data Fig. 2c), which corresponds to a trough in effective

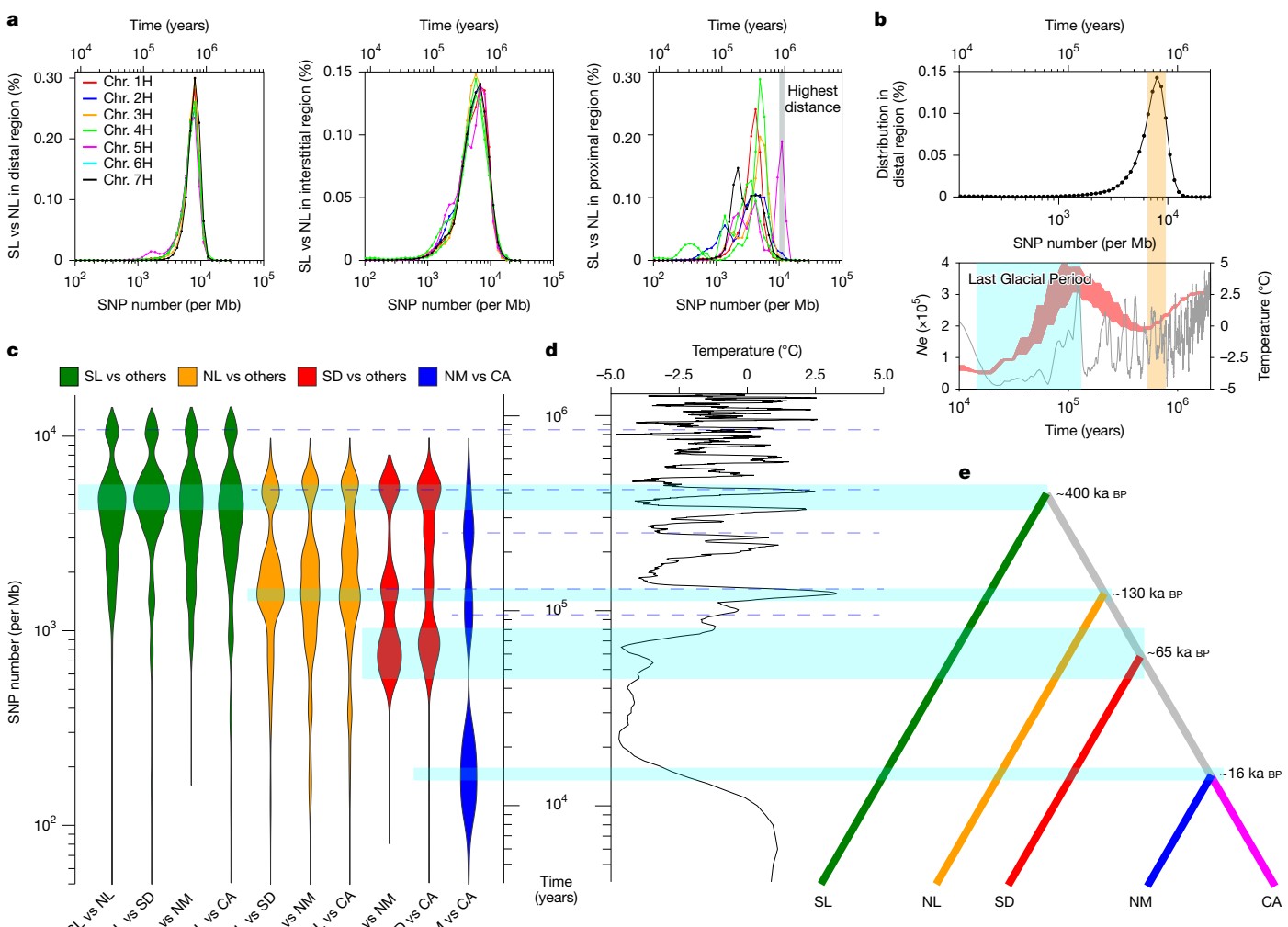

**Fig. 2 | Evolutionary history of wild barley. a**, Distribution of sequence divergence (SNPs per Mb) between pairs of accessions from the SL and NL populations in distal, interstitial and proximal regions. The grey shading in the right panel marks the highest divergence between both populations owing to the presence of deeply diverged haplotypes on chromosome 5H (Chr. 5H). **b**, The distribution of pairwise sequence divergence for all sample pairs in distal regions of the genome (top), and the historic trajectories of effective population sizes in wild barley as inferred by PSMC (red) and global average surface temperatures[39] (bottom). The area chart of population size was based on the sample pairs from the lowest identity-by-state (IBS) bin

(0.6 < IBS < 0.67) in Extended Data Fig. 3b. The orange shading marks a simultaneous decline of population size and temperature that corresponds to peaks in the SNP distribution. The Last Glacial Period (120 to 11 ka BP) is marked by blue shading. *Ne*, effective population size. **c**, Violin plots showing the distributions of pairwise sequence divergence in proximal regions of five wild barley populations. The blue shading highlights the peaks in the distribution that mark the most recent divergence between pairs of populations; earlier such events are marked by dashed lines. **d**, Global average surface temperatures[39] in the past 2 million years. **e**, The divergence of wild barley populations (most recent inferred split times) is represented as a tree.

population size at the same period (Fig. 2b). Fluctuations of population size were also evident from historic trajectories of effective population sizes computed with PSMC[8]. These data indicate that wild barley has recovered from a bottleneck between 2000 and 500 ka BP (Extended Data Fig. 3b). A later bottleneck in all wild barley populations (120 to 11 ka BP) coincided with the Last Glacial Period (Supplementary Fig. 2).

Distributions of divergence time in proximal regions were multimodal and differed between chromosomes (Extended Data Fig. 2c). This observation defies easy explanation. It may stem from the paucity of centromeric haplotypes and their persistence as single linkage blocks on evolutionary timescales. To better understand this pattern, we asked whether the divergence of long centromeric haplotypes reflects the divergence between individuals and the split times between populations (Supplementary Fig. 3). To do so, we used SNPs in pericentromeric regions (centromere ±25 Mb) to calculate pairwise divergence times between wild barley individuals and arranged wild barley populations in a tree structure based on their most recent splits

from each other (Fig. 2c,e and Extended Data Fig. 3a). This representation simplifies the relationships between barley populations. Of note, divergence times were multimodal, and the peaks in the distribution aligned with fluctuations in global surface temperature (Fig. 2d and Extended Data Fig. 3a). This pattern may be attributable to repeated episodes of colonization of new habitats, contraction and potential loss of populations, recolonization and secondary contact between populations. For example, the common ancestor of the Syrian Desert, northern Mesopotamian and Central Asian populations split from the northern Levantine lineage around 120 ka BP when a warm climate may have created new habitats. The northern Mesopotamian and Central Asian populations split around 17 ka BP. This is consistent with the paleoclimatic modelling of Jakob et al.[18], according to whom wild barley was absent from Central Asia as recently as 21 ka BP. The old age of the southern Levantine population (that is, its early divergence from populations elsewhere) is consistent with the supposed status of that region as a glacial refugium[18]. We were intrigued by the presence of a

centromeric haplotype in some southern Levantine wild barleys that diverged from other such haplotypes around 900 ka BP (Fig. 2a and Extended Data Figs. 2c and 4). This is much a deeper split than seen within and between other wild barley populations. The 'relict' haplotype may be a chance escape from genetic drift owing to larger population sizes in the southern Levant or may have been retained by selection for some adaptive advantage it confers. The latter hypothesis is lent some support by the fact that the relict haplotype predominates in many domesticated barley populations (Extended Data Fig. 4d) and that a selective sweep was detected with XP-CLR[22] (Supplementary Fig. 4a). Fang et al.[23] speculated that higher-than-average differentiation between wild barley on chromosome 5H (Extended Data Fig. 1d) may have been caused by a large pericentric inversion on that chromosome. We did see inversions in this region, but they did not extend across the entire haplotypes and occurred in other haplotypes (Extended Data Fig. 4e), making it unlikely that structural variation is the sole explanation for the long persistence of the relict haplotype.

## Haplotype perspective on barley evolution

To add domesticated barley to the picture, we selected from a large collection of 19,778 domesticated barley accessions[24] a panel of 302 samples, of which we sequenced 116 to about tenfold coverage and 186 to about threefold whole-genome coverage (Fig. 1b, Supplementary Tables 1, 2 and 6–8 and Supplementary Figs. 5–8). We ran IntroBlocker on these data. As was observed in wheat, sequence divergence in domesticated barley, in contrast to its wild relative, was bimodal. This was true irrespective of whether distal or proximal regions were considered (Fig. 3a and Extended Data Fig. 5a). The recent peak at around 98 SNPs per Mb (approximately 8,000 years of divergence) corresponds to a bottleneck that marks the coalescence of many haplotypes into common ancestors in the hypothetical domesticated founder population (or populations). The earlier peak (6,500 SNPs per Mb, 530,000 years) mirrors that seen in wild barley and arises from comparisons between haplotypes that diverged before domestication. To group haplotypes according to whether they split before or after domestication, we set a threshold of 400 SNPs per 1-Mb window (corresponding to a divergence time of 32,000 years; Fig. 3a). We give exemplary figures drawn with a 5-Mb window size (Fig. 3b and Extended Data Fig. 5b), but used 100-kb windows after inspecting haplotype length around a key domestication gene (Supplementary Table 9 and Supplementary Fig. 9).

A prominent feature in the whole-genome AHG maps of barley was the presence of long centromeric haplotypes that were shared between wild and domesticated barley. This haplotype sharing lends immediate visual support to the notion of the mosaic ancestry of domesticated barley (Fig. 3b and Extended Data Fig. 5b). Owing to the lower diversity of haplotypes in proximal than in distal regions of the genome in both wild and domesticated barley, our diversity panel covers nearly all pericentromeric haplotypes, but does not achieve saturation in distal regions (Extended Data Fig. 6a). For example, there was only a single pericentromeric haplotype in domesticated barley on chromosome 1H, which was found mainly in Central Asian wild barleys (Fig. 3b). To paint a more general picture, 55.9% of domesticated haplotypes were present in at least one wild barley sample; in the converse scenario, 7.0% of wild haplotypes were shared with a domesticated barley (Fig. 3c). A saturation analysis makes it seem likely that a larger sample of wild genotypes might unearth more shared haplotypes (Fig. 3d). Conversely, some domesticated haplotypes may lack a wild counterpart: haplotypes private to the domesticate tend to be rare (Fig. 3e). They may have arisen after domestication by recombination of haplotypes inherited from the wild progenitors or their progenitors may have been extinct in the wild because of genetic drift. As expected after a bottleneck, the haplotype frequency spectrum differs between wild and domesticated barley. Common haplotypes (that is, those with a major allele frequency above 20%) are seldom seen in wild barley, but were more frequent in

the domesticate (Extended Data Fig. 6b). Still, 79% of haplotypes in domesticated barley with an identifiable wild counterpart occur at low frequency (less than 5%) in the wild (Fig. 3f). Seven regions of the genome showed an extreme reduction of haplotype diversity in domesticated but not in wild barley (Fig. 3g). We inspected local haplotype structure (Supplementary Fig. 10) and annotated the functional effects of genomic variants residing in these intervals to prioritize genes for future inquiry (Supplementary Table 10), even though the large sizes of the regions preclude the confident identification of any single plausible candidate gene. More generally, the high genetic differentiation, evident at the level of both SNPs and haplotypes (Extended Data Fig. 8c,d), may make it impossible to map selection sweeps by outlier scans: in pairwise comparisons between domesticated populations, on average 7.5% of the genome did not share any haplotypes (Supplementary Table 11). Rather than from pervasive forces of adaptive evolution, we suspected that local lineage sorting may underlie this pattern.

## Spatiotemporal origins of haplotypes

We enquired into the temporal and spatial origin of haplotypes in domesticated barley by running IntroBlocker with different thresholds corresponding to divergence time brackets and inspecting which extant wild barley genomes have the closest relatives of domesticated barley (Fig. 4 and Extended Data Fig. 7). The resultant genome map of spatiotemporal relations is again testimony to the mosaic genomic constitution of the crop (Fig. 4a and Extended Data Fig. 7). The mosaic structure appears to have emerged early in the evolution of cultivated barley. About 91% of domesticated haplotypes with a wild counterpart split from the latter between 32 and 8 ka BP, that is, during the formation of the immediate wild progenitor of domesticated barley and the initial stages of domestication (Fig. 4b). Fewer than 9% are attributable to more recent gene flow. All five wild barley populations contributed to domesticated barley, albeit in different proportions. Wild barley populations from the southern and northern Levant and Central Asia each contributed between 20% and 27% of haplotypes and those from the Syrian Desert and northern Mesopotamia contributed 16.4% and 12.9%, respectively (Fig. 4b). There were also differences between domesticated barleys as to how much certain wild populations contributed genetic material to them. Haplotypes from Central Asian wild barleys were found more frequently in domesticated barleys from East and Central Asia than in other domesticated populations (Fig. 4d). This close affinity between wild and domesticated barley from 'the East' had been noted by Morrell et al.[25], who saw it as evidence for a second centre of barley domestication east of the Zagros mountains in Iran. Our explanation is that this trend occurred due to gene flow from local wild populations into already domesticated populations coming from the western Fertile Crescent. The northern Levantine wild barley population contributed more to domesticated forms in western Asia and Europe than to those in East and Central Asia, which had more Central Asian ancestry. Mediterranean barleys had a higher share of southern Levantine ancestry. This relationship may suggest different points of departure of early farmers from the Fertile Crescent. These results are qualitatively similar to those of Poets et al.[2], but differ in that their analysis, based on 5,000 SNP markers, assigned a greater contribution (more than 50%) of southern Levantine wild barley to all domesticated populations.

Domesticated barleys differ also in how much recent gene flow they have received from wild barley (Fig. 4c,e). Wild introgressions are most common in cultivated accessions from western and Central Asia, where wild barlEy is common: 12.8% of haplotypes in Syrian barleys (SYR-THM) are attributable to recent (later than 8 ka BP) wild introgressions (mainly from the Central Asian and northern Mesopotamian populations). We were surprised to see wild haplotypes flowing into northern European barley in apparently recent times: the cultivar 'Kiruna' (HOR_17134) shared a haplotype on chromosome 7H,

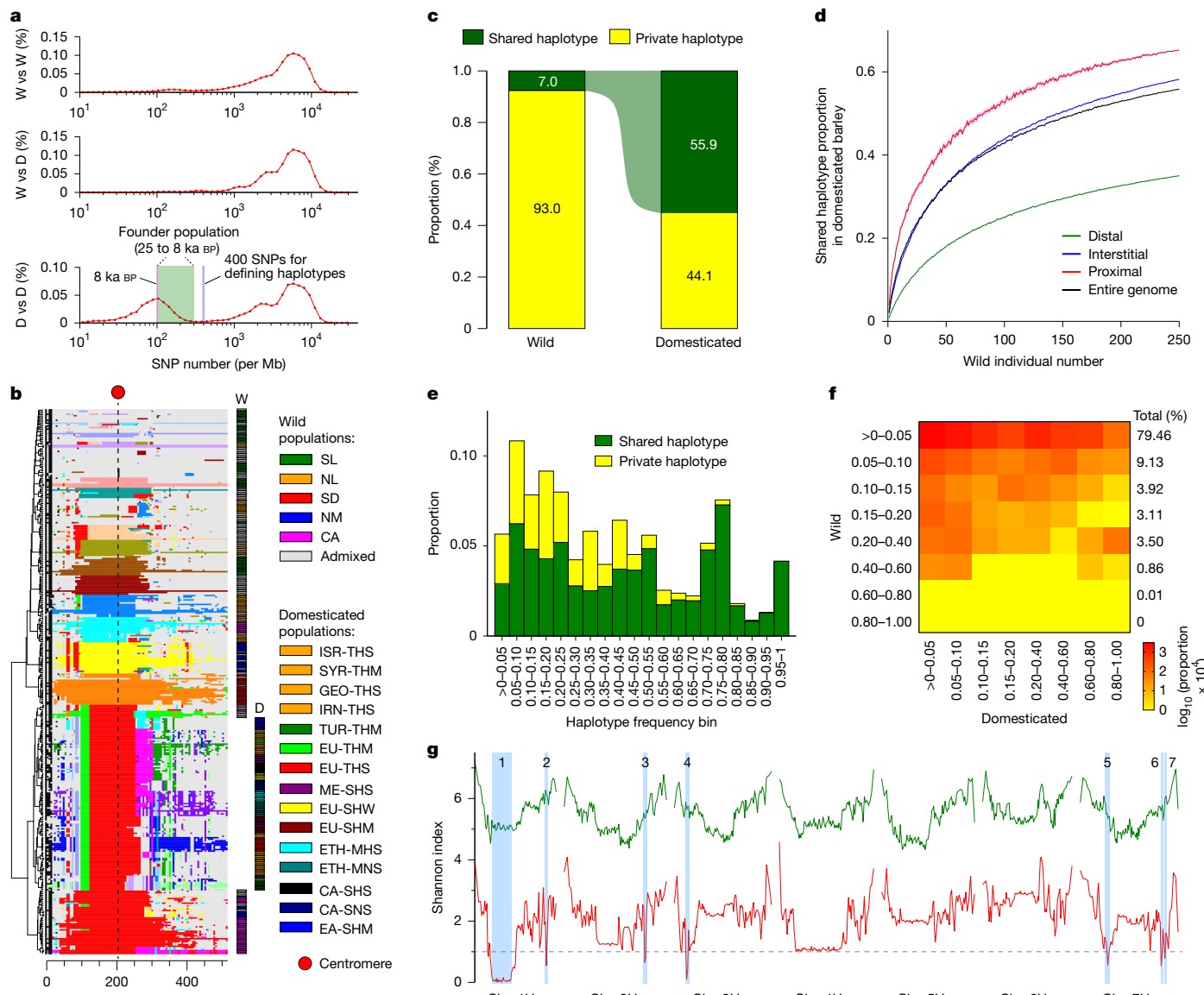

**Fig. 3 | Haplotype diversity in wild and domesticated barley. a**, Sequence divergence (SNPs per Mb) between pairs of wild (W) and domesticated (D) barley. The blue line marks the 400 SNPs per Mb threshold used by IntroBlocker to distinguish pre-domestication and post-domestication haplotypes. The green shading indicates the persistence of a founder population that began fragmenting into isolated groups from 8 ka BP (purple line). The recent peak started at approximately 300 SNPs corresponds to 25 ka BP, marking divergence between ancestral domesticated and present-day wild barley. **b**, AHGs on chromosome 1H inferred using 5-Mb windows. The 20 most frequent AHGs are shown in colour; grey indicates rarer AHGs. Sample groups (wild or domesticated) are indicated by side bars. **c**, Proportions of shared versus private haplotypes in wild and domesticated barley. **d**, Saturation curves showing how the proportion of shared haplotypes increases with the number of wild barley samples. The lines and shading denote the average and 95% confidence intervals, respectively, from 100 random subsamples.

**e**, Proportions of haplotypes in domesticated barley that are shared with wild barley, grouped by their frequency in the domesticated barley and divided into 20 equal intervals (for example, 0–0.05, 0.05–0.1,..., 0.95–1.0). The data represent the genome-wide pattern across all seven barley chromosomes. **f**, Normalized two-dimensional haplotype frequency spectrum in wild and domesticated barley. Each cell shows the proportion of shared haplotypes (cell count divided by total counts) displayed on a log scale after multiplying by $10^4$. The percentages on the right margins indicate the relative sizes of frequency bins in domesticated barley (row sums), for example, 79.46% of shared haplotypes occurs at 5% frequency or less in wild barley. **g**, Haplotype-based Shannon diversity indices in wild (green line) and domesticated (red line) barley populations. Seven regions (blue shading) with an index of less than 1 in domesticated barley (dashed line) were defined as putative selective sweeps. The reference genome used was B1K-04-12.

100–200 Mb with a Central Asian wild barley (Fig. 4f,g and Supplementary Fig. 11). This observation can be explained by the use of wild barley as a genetic resource by breeders: Kiruna's pedigree features 'Vogelsanger Gold', a variety from the 1960s with a wild barley introgression[26]. The same haplotype is seen in HOR_17572, which is purported to be an Austrian landrace. We considered errors in the passport records or accidental outcrossing during ex situ management as the most likely explanation for this case.

## Domesticated lineage relationships

We inspected divergence levels between haplotypes post-domestication to infer split times between different populations of domesticated barley in a hierarchical manner (Fig. 5a,b, Extended Data Fig. 8a,b and Supplementary Table 12). We used only SNPs in haplotypes descended from the same wild lineage to compute pairwise divergence times between samples. First, we divided our domesticated barley panel

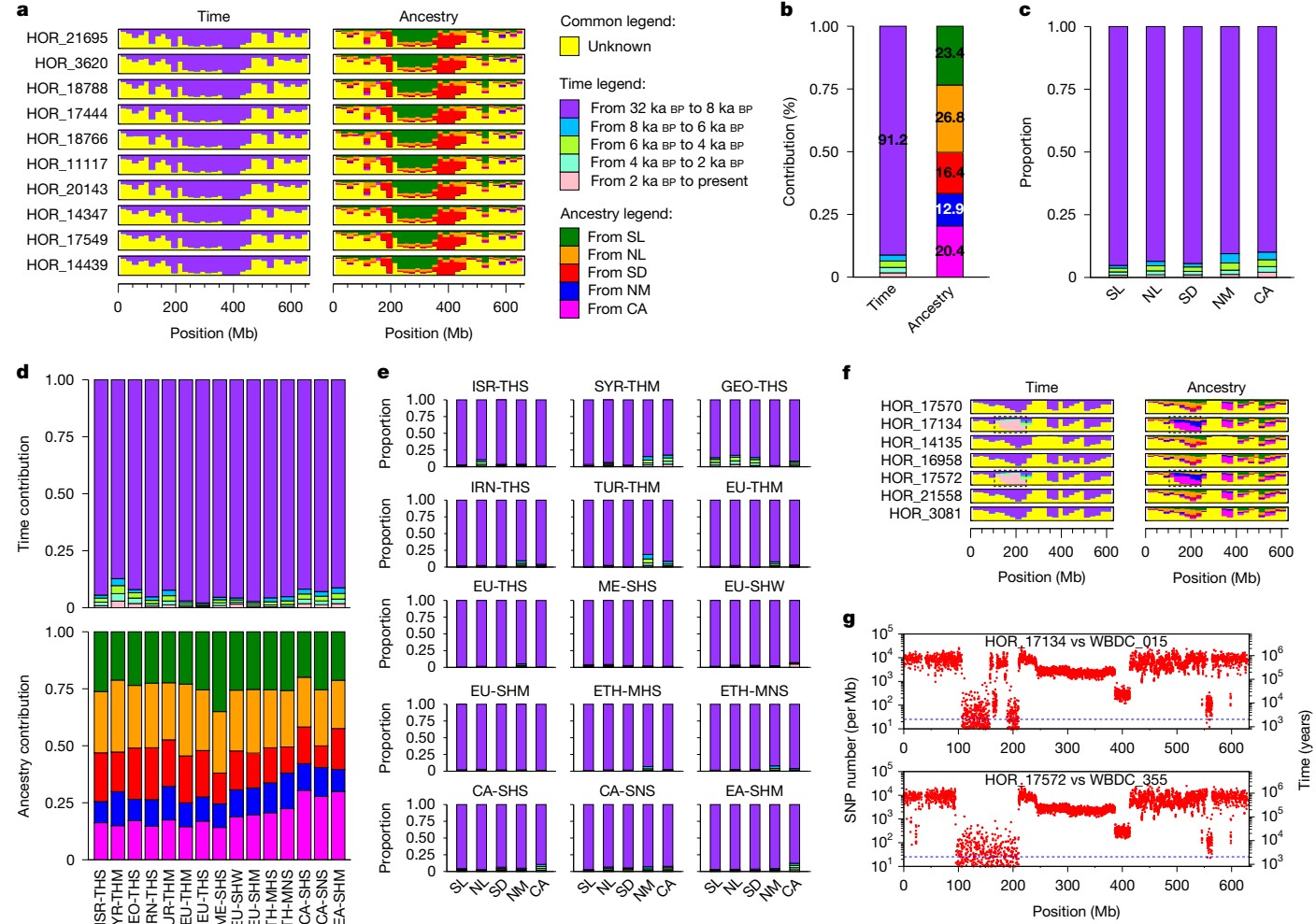

**Fig. 4 | Spatial and temporal origins of haplotypes in domesticated barley.**
**a**, Spatiotemporal origins of haplotypes of the EU-THS population of
domesticated barley on chromosome 2H in 20-Mb windows. The height of the
coloured bars is proportional to the probability that a haplotype entered the
domesticated gene pool in a certain time period (left) or from a certain wild
barley population (right). Yellow denotes haplotypes of unknown provenance
owing to missing data, lack of a clear wild counterpart or potential gene flow
from domesticated to wild barley. The results for all domesticated populations
are shown in Extended Data Fig. 7. **b**, Spatiotemporal origins of domesticated
barley haplotypes across the entire genome. **c**, Time periods at which
haplotypes from each of the five populations entered the domesticated

gene pool. **d**, Spatiotemporal origins of haplotypes in 15 domesticated barley
populations. **e**, Time periods at which haplotypes from each of the five
populations entered 15 domesticated barley populations. Haplotypes of
unknown provenance were ignored when considering proportions in panels **b**–**e**.
**f**, Spatiotemporal origins of haplotypes of the EU-SHW population on
chromosome 7H. The dashed rectangle marks a haplotype that owes its
presence in domesticated barley to recent gene flow. **g**, Sequence divergence
(SNPs per Mb) on chromosome 7H between two EU-SHW accessions
(HOR_17134 and HOR_17572) and two Central Asian wild barleys (WBDC_055
and WBDC_355). The dashed line marks 2 ka BP of divergence (random mutation
rate of 6.13 SNPs per Gb per generation).

into three groups: western (Near East + Europe), eastern and Ethio-
pian barleys, which all diverged from each other around 8.5 ka BP,
reflecting the dispersal of agriculture from the Fertile Crescent
around that time. Subsequently, western barley split into three lin-
eages (Near East, two-rowed Europe and six-rowed Europe) around
7.5 ka BP. This is consistent with the archaeological records that show
that by 7 ka BP barley had been introduced to Europe, North Africa
and Central Asia[27]. These populations subdivided further between
7 and 5 ka BP. Divergence time distributions had multiple peaks
in some comparisons. In the case of European barleys, gene flow
between populations, which are differentiated by morphology and
phenology rather than by geography, is plausible. In the case of west-
ern Asian populations from Georgia (GEO-THS) and Iran (IRN-THS),
fine-scale population is conceivable: landraces in these mountain-
ous regions may trace back to a common source population but have
evolved in mutual reproductive isolation after reaching their current
habitats. In Fig. 5c, we provide a graphical summary of these results

in relation to known dispersal routes supported by archaeological
evidence[27].

## A single-gene view of mosaic ancestry

How we think about barley crop evolution owes much to the genetic
dissection of loci at which mutant alleles confer traits that are seen only
in the domesticate, namely, non-shattering ('non-brittle') spikes, fertile
lateral grains ('six-rowed' spikes) and the loss of lemma adherence to the
mature grain ('naked' or 'hulless' barley). The corresponding genetic
loci are *BRITTLE RACHIS 1 and 2* (ref. 6), *SIX ROWED SPIKE 1* (ref. 28)
and *NUDUM*[29] with mutant alleles *btr1*, *btr2*, *vrs1.a1*–*vrs1.a4* and *nud*.
These genes were not identified in genome-wide scans for regions
with extraordinarily low haplotype diversity (Extended Data Fig. 6c,d).
The reason for this is that multiple independent loss of function of
alleles are present at the *BTR1/2* and *VRS1* loci and that the widespread
cultivation of naked barleys is confined to a few geographical regions

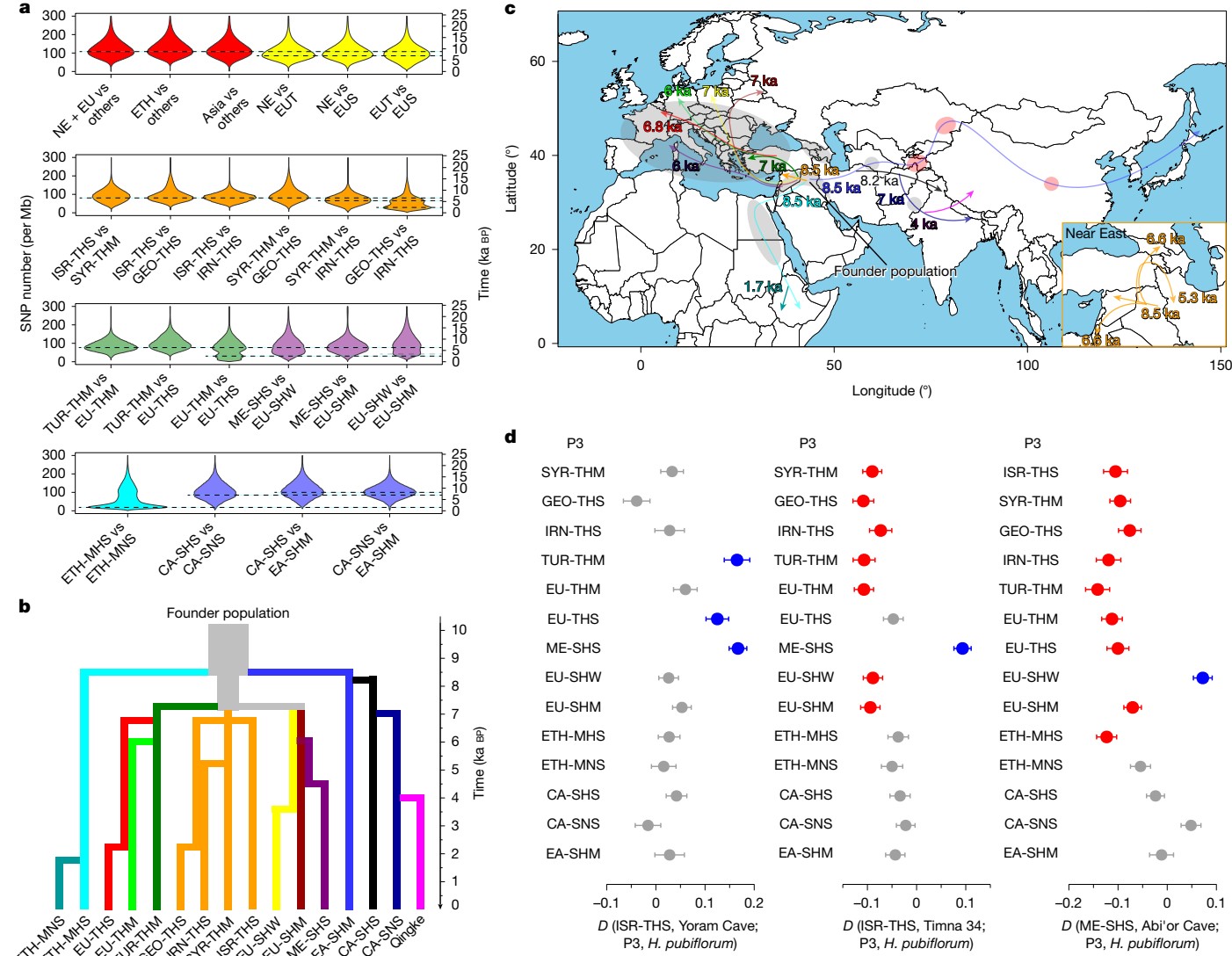

**Fig. 5 | Divergence and dispersal of domesticated barley. a**, Violin plots showing the distribution of sequence divergence (SNPs per Mb) in pairwise comparisons between samples from different populations of domesticated barley. The dashed lines mark the peaks of the distributions (split times). Multimodal distributions may have risen from episodes of gene flow. **b**, Schematic illustrating the lineal descent and split times between 15 barley populations defined in this study and Tibetan barleys (Qingke) studied by Zeng et al.[40]. **c**, Map showing when and along which routes domesticated barley spread from its centre of origin in the Fertile Crescent. The grey shading indicates barley archaeological sites dating back about 7,000 years; the red shading indicates barley archaeological sites dating back about 5,000 years[27]. We took archaeological sites[27], divergence time and population structure

(Supplementary Fig. 8) into account when drawing this figure. Geographical outlines were obtained from the R package 'maps' (https://CRAN.R-project.org/package=maps), which uses public domain base map data (under a GNU General public license: version 2). **d**, *D* statistics for different comparisons among ancient barleys and 15 domesticated barley populations. The outgroup was *H. pubiflorum*. Blue and red indicate significant results (|*Z* score| > 3), whereas grey indicates non-significant results (|*Z* score| < 3). A positive *D* value (blue) suggests gene flow between P1 and P3, whereas a negative *D* value (red) indicates gene flow between P2 and P3. The grey bars, with *D* values close to zero, imply no detectable gene flow. The solid circles represent *D* values. The error bars indicate ±1 standard error. Sample sizes of each population, standard deviations and *Z* values are provided in Supplementary Table 19.

such the Himalayas and Ethiopian highland. Even so, the persistence of long haplotypes (Extended Data Fig. 9a–c and Supplementary Fig. 12) around these genes and the accumulation in them of rare variants since the most recent, and indeed recent (less than 10 ka), common ancestor allowed us to date, in an approximate manner, the origin of domesticated loss-of-function alleles. We used Genealogical Estimation of Variant Age (GEVA)[30] to estimate ages of mutant alleles and their surrounding haplotypes at the *Btr1/2*, *Vrs1* and *Nud* loci. Our estimated age of 27 ka BP for the *btr1* haplotype (Extended Data Fig. 9d) predates the earliest archaeobotanical remains of domesticated barley by some 17,000 years[1,5], but it is closer to the approximately 22 ka BP estimate from an archaeobotanical modelling study[31]. It is not impossible that non-shattering barleys (and the causal haplotypes) languished as rare

variants in the wild before early cultivators selected them for propagation. The *btr2* haplotype originated around 15 ka BP, which is very close to the approximately 12 ka BP estimate from Allaby et al.[31]. The *vrs1.a1* haplotype dates back to approximately 25 ka BP, consistent with the identification of it as the most ancient six-rowed allele[28]. By contrast, *vrs1.a2* emerged around 8 ka BP, in line with the hypothesis that it was derived from cultivated two-rowed barley[28]. The age estimation for *vrs1.a4* (approximately 7 ka BP) matches its current geographical distribution, being limited to Central and East Asian cultivated barley[32]. As a control, we also estimated the ages of functional (dominant) haplotypes of the *Vrs1* and *Nud* genes. The estimated origins of the *Vrs1.b2*, *Vrs1.b3* and *Nud* haplotypes far predate domestication − 26 ka BP, 55 ka BP and 35 ka BP, respectively. As a further control, randomly selected

wild barley SNPs from the same genomic regions show estimated ages between 100 and 120 ka BP (Supplementary Fig. 13), consistent with the peak of effective population size inferred from the PSMC results (Fig. 2b). The *ppd-H1* haplotype, which confers photoperiod insensitivity[33], is estimated to be approximately 30 ka BP in age, which supports the view that it originated before domestication[34]. Even though the precision of molecular dating is limited by uncertainties surrounding mutation rate estimates, we can propose the following relative order of emergence of mutant alleles and their surrounding haplotypes: *btr1*, *vrs1.a1*, *nud*, *btr2*, *vrs1.a3*, *vrs1.a2* and *vrs1.a4* (Extended Data Fig. 9d). Their most closely related wild counterparts (Extended Data Fig. 9e,f) were found in different present-day wild barley populations: southern Levant (*btr1*, *nud* and *vrs1.a3*), northern Levant (*btr2* and *vrs1.a2*) and northern Mesopotamia and Central Asia (*vrs1.a1* and *vrs1.a4*). This result aligns with earlier gene-based analyses of the *btr1/btr2* locus by Pourkheirandish et al.[6], who posited two origins of tough-rachis barleys, one in the northern and the other in the southern Levant. The early origin of the *nud* mutation (16 ka BP) is consistent with the fact that hulless barleys from places as far apart as Tibet and Ethiopia all share the same 17-kb deletion spanning the *NUD* genes (Extended Data Fig. 9c). Yet, their overall genomic composition is quite different: the ETH-MNS and CA-SNS population do not share any haplotypes in 44.8% of the genome. We speculate that before the respective ancestors of Central Asian and Ethiopian barleys left the Fertile Crescent, they acquired the common *nud* allele as it was spreading from a single southern Levantine source across barley's early gene pool.

## Ancient DNA reveals persistent structure

We analysed ancient DNA sequences of 23 barley grains (Fig. 1c and Supplementary Table 16) dated to between 6000 and 2000 calibrated years before present (cal BP) to see how they might complement our haplotype map of extant genomes. Fragment lengths, nucleotide misincorporation profiles and high mapping rates (Supplementary Table 16 and Supplementary Fig. 14) confirmed authenticity. All ancient barleys grouped together with cultivated types in a PCA (Extended Data Fig. 10a) and had the domesticated *btr1Btr2* haplotype, common in western barleys (Supplementary Table 16). The barleys from Yoram Cave and Timna 34 were two-rowed forms with the *Vrs1.b2* allele, likewise common in western types (Supplementary Table 16). Those from Abi'or Cave carried the six-rowed (*vrs1.a1*) allele. In most cases, the ancient barley samples carried the same long pericentric haplotypes as modern domesticated barley, with only a few exceptions on chromosome 7H, where some ancient barleys contained private haplotypes (Extended Data Fig. 10b and Supplementary Fig. 15). These analyses indicate that the ancient genomes derive not from direct descent of local wild stands but from a more widespread founder population that gave rise to cultivated barley across the Fertile Crescent.

We used identity-by-state clustering with our 19,778-sample genotype-by-sequencing (GBS) panel (Supplementary Table 17 and Supplementary Fig. 16) and genome-wide SNP-based phylogenetic trees (Extended Data Fig. 10c and Supplementary Fig. 17) to understand the relationship between our ancient samples and present-day barley populations. Both analyses supported the clustering of the two-rowed Yoram Cave and Timna 34 samples with the modern ISR-THS population, whereas the six-rowed Abi'or Cave sample clusters with the ME-SHS population. As the grains from Abi'or Cave were dated to 2000 cal BP, that is, to the Roman period, secondary contact between geographically distant barley population may have been mediated by sea-borne trade across the Mediterranean. Owing to the limited number of high-coverage ancient samples (only two per archaeological site), population-level assessments of genetic diversity were not feasible. We sought to detect changes in genetic diversity from ancient to modern Israel barley at the single-sample level by comparing the number of rare alleles (present in wild barley) found in individual ancient and modern genomes (Supplementary Table 18 and Supplementary Fig. 18). The results revealed a gradient in diversity, with Yoram Cave (6 ka BP) showing the lowest, followed by ISR-THS and Timna 34 (3 ka BP) with similar levels, and Abi'or Cave (2 ka BP) exhibiting the highest diversity (Extended Data Fig. 10d and Supplementary Fig. 19). Generally, ancient samples tended to show higher genetic diversity than modern samples. This is because, over the course of prolonged domestication, selective breeding and modern agricultural practices, the gene pools of crops and livestock have often experienced bottlenecks and strong selective pressures, leading to a reduction in genetic diversity. To explain why ancient barleys from Israel shows increasing genetic diversity in more recent times, we used *D* statistics to test for gene flow that might have caused this pattern (Fig. 5d and Supplementary Table 19). We performed *D* (ISR-THS, Yoram Cave; P3, *H. pubiflorum*), *D* (ISR-THS, Timna 34; P3, *H. pubiflorum*) and *D* (ME-SHS, Abi'or Cave; P3, *H. pubiflorum*). These tests found no detectable gene flow in the Yoram Cave samples, but reveal significant western introgression into Timna 34, and even stronger gene flow into samples from Abi'or Cave. These findings align with the genetic diversity patterns, further supporting the observed gradient. The gene flow detected between Israel and western barley populations approximately 3000 years ago could plausibly be attributed to interregional trade or population movements during the Late Bronze Age. Such exchanges may have involved the intentional or unintentional transport of barley grains or seeds, contributing to the observed genomic admixture between geographically distant regions. For modern two-rowed Israel domesticated barley (ISR-THS), sampling information indicates that they are admixed samples (Supplementary Table 7), and *D* statistics show that they still exhibit gene flow from the Mediterranean population (ME-SHS). Our study expands the understanding of the historical dynamics of modern Israel barley populations. Domestication-related selection may not have been the primary driver of diversity changes in Israel barley. Instead, long-distance trade, human migration and associated gene flow appear to have had a substantial role in enhancing the genetic diversity of modern cultivated barley in the region.

## Discussion

Our findings support and extend the two previously proposed models for the mosaic ancestry of domesticated barley, as outlined by Pankin et al.[3]: (1) recurrent introgressions from diverse wild populations into an early domesticated 'proto-vulgare' lineage, and (2) a pre-existing population structure within the wild progenitor gene pool. These mechanisms are not mutually exclusive, and our data suggest that both shaped the genomic composition of domesticated barley. The haplotypic origins of key domestication loci — *Btr1/Btr2*, *Vrs1* and *Nud* — underscore this complexity. Their geographically disparate wild relatives point to a polycentric domestication phase, followed by a protracted period of cultivation localized within the Fertile Crescent (Extended Data Fig. 11). During this time, regionally structured barley populations — and the human communities that managed them — began to diverge genetically while maintaining some connectivity.

Our divergence-time analyses further support this scenario. The recent peak in haplotype divergence beginning at approximately 300 SNPs (Fig. 3a), corresponding to approximately 25 ka BP, marks the emergence of a distinct genetic lineage leading to domesticated barley. This estimate aligns with PSMC-inferred declines in effective population sizes between 25 and 10 ka BP (Supplementary Fig. 20), reinforcing the view of a protracted 'proto-domestication' phase. This timing is also consistent with archaeobotanical evidence: the high frequency (approximately 36%) of domestic-type abscission scars in wild barley at the Ohalo II site (23 ka BP)[35] suggests that cultivation predated the fixation of canonical domestication traits such as non-brittle spikes[35,36].

Gene flow between early cultivated populations and nearby wild barleys contributed to the observed mosaic ancestry. In western Asia, this ongoing introgression continued well beyond domestication, as reflected in modern haplotype sharing patterns. Consequently, present-day western Asian barley varieties are unlikely to be direct descendants of the original domesticated founder population. As agriculture expanded beyond the Fertile Crescent, domesticated barley diversified into regionally distinct lineages. This geographical spread, accompanied by a decline in wild introgression, resulted in populations evolving largely in isolation. However, geography alone does not explain the observed structure. Agricultural practices and selection for distinct agronomic traits — such as the divergence between European two-rowed and six-rowed barleys — further shaped gene pools.

This evolutionary process has important implications for functional genomics. The deep haplotype differentiation across populations — arising from both ancestral structure and post-domestication gene flow — may confound signals of adaptation. In many genomic regions, different lineages carry no shared haplotypes, a pattern also observed in selective sweeps. This overlap between mosaic ancestry and selection signatures complicates the mapping of adaptive loci. One promising avenue for disentangling these effects is mutational genomics, where causative variants can be traced across structural and geographical contexts. The example of *HvCENTRORADIALIS* illustrates this approach: initially identified in classical barley mutant as a major flowering-time regulator, its broader role became apparent only through population sequencing, which revealed both structural variation[37] and association with range expansion[38].

In summary, our haplotype-based analysis provides a high-resolution view of barley domestication and post-domestication evolution. It supports a model in which early cultivation involved multiple wild sources, followed by gene flow, geographical divergence and local adaptation. This complex legacy continues to shape the genomic architecture of modern barley, and poses both challenges and opportunities for future genetic and breeding research.

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

# Methods

## Sample selection for genome sequencing

**Wild barley.** Our wild barley panel (Supplementary Table 1) comprised 285 accessions from the Wild Barley Diversity Collection (WBDC)[41,42], a collection of ecogeographically diverse accessions. The whole-genome sequencing (WGS) of the WBDC collection has been described in a companion paper[17]. A further 95 diverse barley accessions, mainly from the panel of Russell et al.[14], were also included. The latter set of samples had been sequenced to approximately 3× coverage by Jayakodi et al.[37]. In the present study, we resequenced 32 of these samples to increase their coverage to approximately 10×.

**Domesticated barley.** Milner et al.[24] defined 12 populations using model-based ancestry estimation with ADMIXTURE[43] in a global diversity panel of 19,778 domesticated barley, which had been subjected to GBS[24]. We used the ADMIXTURE results and GBS SNP matrix of Milner et al.[24] for sample selection. Except for the Near-eastern population (coloured orange in figure 1b of Milner et al.[24]), we selected samples according to the following procedure. First, unadmixed samples, that is, those with an ADMIXTURE ancestry coefficient $q \geq 0.95$ were used as input for a PCA with smartpca[44] (v7.2.1). Then, samples were selected to cover the PCA space evenly (Supplementary Fig. 5). Owing to its higher genetic diversity and internal substructure, a more sophisticated procedure was followed for the Near-eastern population (Supplementary Table 7 and Supplementary Fig. 6). First, ADMIXTURE[43] (v1.23) was run on 1,078 samples of Milner et al.[24], where the Near-eastern ancestry coefficient $q$ was higher than that of all other populations, with $q$ ranging from 0.25 to 0.98. Before running ADMIXTURE, the SNP set was thinned with PLINK[45] (v1.9) using the parameters '--indep-pairwise 50 10 0.1'. For each value of $K$ (the number of ancestral populations) from 2 to 6, the output of 15 replicate runs of ADMIXTURE with different random seeds was combined with CLUMPP[46] (v1.1.2) and plotted with Distruct[47] (v1.1). Individuals with $q \geq 80\%$ for their main ancestry component were considered unadmixed. The results for $K = 6$ was chosen for further analysis. The genetic separation of the defined populations was confirmed with smartpca[44] (v7.2.1). Only those samples of the Near-eastern subpopulations that were actually located in the Near East were selected for sequencing. The selected samples were distributed in an equidistant manner in the PCA diversity space. In total, we selected 302 samples from 15 populations (Supplementary Table 8 and Supplementary Fig. 7). The populations were named according to their geographical origins and three key traits closely connected to global population structure[24] (Supplementary Table 6): row type (two-rowed (T), six-rowed (S) and mixed (M)); lemma adherence (hulled (H) and naked (N)); and annual growth habit (winter sown (W), spring sown (S) and mixed (M)). For example, ISR-THS refers to a population whose members are predominantly two-rowed hulled spring barleys from Israel. For each population, we selected about 20 accessions for WGS sequencing. Among these, 7–10 samples of each population (total: 116, 'high-coverage samples') were sequenced to approximately tenfold coverage. The remaining samples of each population were sequenced to approximately threefold coverage (total: 186, 'low-coverage samples'). Seeds for all selected accessions can be ordered from the German Federal ex situ genebank at IPK Gatersleben.

## Plant growth, DNA isolation and Illumina sequencing

Plant cultivation and DNA isolation were essentially as previously described[24]. Illumina Nextera DNA Flex WGS libraries were prepared and sequenced (paired end: 2 × 151 cycles) on an Illumina NovaSeq 6000 device at IPK Gatersleben according to the manufacturer's instructions (Illumina).

## Reads mapping and variant calling

The reads of 682 barley genotypes, of which 380 were wild and 302 domesticated, were mapped to the MorexV3 genome sequence assembly[15] using Minimap2 (v2.24)[48]. Mapping statistics of all 682 accessions are shown in Supplementary Table 1. BAM files were sorted and deduplicated with Novosort (v3.06.05; https://www.novocraft.com/products/novosort/). Variant calling was done with bcftools (v1.15.1)[49] using the command 'mpileup -a DP,AD -q 20 -Q 20 --ns 3332'. The resultant 'raw' SNP matrix was filtered as follows: (1) only biallelic SNP sites were kept; and (2) genotype calls were deemed successful if their read depth $\geq 2$ and read depth $\leq 50$; otherwise genotypes were set to missing. SNP sites with fewer than 20% missing calls, and fewer than 20% heterozygous calls were used for ADMIXTURE runs (with $K$ ranging from 2 to 4) as described above. At $K = 4$, wild individuals with 15% or more ancestry from domesticated barley were considered admixed. A total of 80 wild admixed samples were excluded from subsequent analyses (Supplementary Table 2 and Supplementary Fig. 21). A total of 251 wild barley samples with high coverage (approximately 10×) without domesticated admixture were used for subsequent population genetic analyses (Supplementary Table 3).

We prepared two SNP matrices, SNP1 and SNP2, for downstream analysis. For SNP1, we extracted the data for 367 (251 wild and 116 domesticated) high-coverage samples from the raw SNP matrix. SNP1 was filtered as follows: (1) only biallelic SNP sites were kept; (2) homozygous calls were deemed successful if their read depth $\geq 2$ and read depth $\leq 50$ and set to missing otherwise; and (3) heterozygous calls were deemed successful if the allelic depth of both alleles was 5 or more and set to missing otherwise. The SNP2 matrix contained variants for 302 domesticated samples and was constructed from another bcftools run using the same parameters as above but with a downsampled dataset, in which the read alignments of the high-coverage samples ($n = 116$) had been thinned so as to achieve a sequence depth comparable with that of the low-coverage samples (Supplementary Fig. 22) using SAMtools (v1.16.1)[49] with the command 'samtools view -s 0.FRAC' (FRAC is the sampling rate). The targeted number of uniquely mapped (Q20), deduplicated mapped reads for the downsampled high-coverage data was set to a random number between 35 million and 52 million. Note that the read length was 2 × 150 bp in all samples. The matrix SNP2 was filtered as follows: (1) only biallelic SNP sites were kept; (2) homozygous calls were considered successful if their read depth $\geq 2$ and read depth $\leq 20$ and set to missing otherwise; and (3) all heterozygous calls were set to missing. A flow chart describing the construction of the SNP matrices used in this study is shown in Supplementary Fig. 23. In analyses in which the use of an outgroup was required, we used WGS data of *Hordeum pubiflorum*[50]. Read mapping and SNP calling were done as described above with one difference: a VCF file for all sites in the genome, including those identical to the reference genome, was obtained. This VCF file was merged with other VCF files to determine ancestral states. We also prepared a SNP matrix with 367 (251 wild and 116 domesticated) high-coverage samples using B1K-04-02 (FT11) as the genome reference[16] for candidate gene search and SNP age calculation. The reads mapping, SNP calling and filtering procedures were the same as those used for generating the SNP1 matrix.

## SNP-based genetic distances

The number of SNPs between any two high-coverage genotypes were calculated as follows. First, pairwise SNP numbers were determined in genomic windows with PLINK2 (v2.00a3.3LM)[51] with the command 'plink2 --from-bp x --to-bp y --sample-diff counts-only counts-cols=ibs0,ibs1 ids=s1 s2 …', where $x$ and $y$ are the start and end coordinates of a window and 's1 s2 …' is a list of sample IDs. Different window sizes were used: 100 kb (shift of 20 kb), 500 kb (shift of 100 kb), 1 Mb (shift of 200 kb), 2 Mb (shift of 400 kb) and 5 Mb (shift of 5 Mb). Then, in each window, a normalized distance measure was calculated to account for the fact that owing to differences in the mappability of short reads, the effective coverage differs between genomic windows[40] (Supplementary Fig. 24). Per-bp read depth was determined for each

sample and each position of the reference genome with the command 'samtools view -q 20 -F 3332 | samtools depth'. The effectively covered region of each window was defined as the union of sites with read depths between 2 and 50. For each, pairwise comparison between samples, the effectively covered regions were intersected using a Perl script. The pairwise distance in a genomic window was calculated as (hom + het/2)/cov, where hom and het are numbers of homozygous and heterozygous differences, respectively, and cov is the size of the intersection of the effectively covered regions of both samples. Genomic windows were considered only if the latter quantity amounted to half the size of the window; otherwise the distance was set to missing.

## Validation of SNP number estimation using accurate long reads
To evaluate the accuracy of our SNP number estimates, we used data from the second version of the barley pangenome[16]. Among the 76 accessions included in the barley pangenome, 13 overlapped with our sample set (Supplementary Table 20). We downloaded the HiFi reads of these 13 accessions and aligned them to the MorexV3 reference genome[15] using pbmm2 (v1.10.0; https://github.com/PacificBiosciences/pbmm2). For HiFi reads, the effectively covered region was defined in the same manner as above but with read depths between 10 and 50 considering average HiFi sequencing coverage of approximately 25×. Variant calling was performed with DeepVariant (v1.6.0)[52] to generate GVCF files for each sample, followed by joint genotyping using GLnexus (v1.3.1)[53,54] to obtain a SNP matrix across the 13 samples. We applied the following filtering criteria: (1) only biallelic SNPs were retained; (2) only genotype calls with depth between 10× and 50× were kept; otherwise, the genotype was set as missing; and (3) for heterozygous calls, we required a minimum allele depth of 10 for each allele. We compared the effective covered region (uniquely mapped regions) of short-read and HiFi-read data across these 13 samples, as well as the intersection of effective covered regions between each pair of samples (Supplementary Fig. 25). The missing rate was calculated for each sample as the number of missing genotype calls divided by its effectively covered region. We then calculated pairwise SNP number between samples using the same method as described above, with a window size of 1 Mb (shift of 200 kb). Only 1-Mb windows in which the intersection of effective covered regions between the two samples exceeds 0.5 Mb were retained for SNP number calculation. Given that SNP number distributions along chromosomes are not always normally distributed — and may even be bimodal in certain cases — we applied Kendall rank correlation to evaluate the consistency between SNP numbers calculated from short reads and HiFi reads (Supplementary Figs. 26 and 27). Confidence intervals for Kendall's tau correlation coefficients were calculated using a percentile bootstrap method with 1,000 resamples.

## Linkage disequilibrium decay
The barley genome was split into three compartments (distal, interstitial and proximal) based on recombination rates[19] (Supplementary Table 21 and Supplementary Fig. 28). Linkage disequilibrium decay was calculated for both wild and domesticated barley in each compartment using PopLDdecay (v3.42)[55] with the command '-Het 0.99 -Miss 0.2 -MAF 0.01 -MaxDist 500'.

## Population structure and divergence times in wild barley
Variants calls of 251 high-coverage wild barley samples were extracted from the matrix SNP1 (see above). SNP sites with fewer than 20% missing calls, fewer than 20% heterozygous calls and minor allele frequency (MAF) ≥ 5% were used in population structure analysis. Model-based ancestry estimation was done with ADMIXTURE (v1.23)[43]. The number of ancestral populations $K$ ranged from 2 to 5. At $K = 5$, individuals with more than 85% of its main ancestry were considered as unadmixed wild barleys. PCA was done with smartpca (v7.2.1)[44]. Genotype calls of the outgroup sample $H.$ $pubiflorum$ were merged with the SNP matrix, and

an IBS-based genetic distance matrix was calculated with PLINK (v1.9)[45]. The distance matrix was used to construct a neighbour-joining tree with Fneighbor (https://emboss.sourceforge.net/apps/cvs/embassy/phylipnew/fneighbor.html), which is part of the EMBOSS package[56]. The resultant tree was visualized with Interactive Tree Of Life (iTOL; v7)[57]. In each of the five wild barley subpopulations, the nucleotide diversity[58] (π) and Watterson's estimator[59] ($\theta_W$) were calculated from the SNP matrix without MAF filtering using a published Perl script[40]. Pairwise fixation indices ($F_{ST}$) between pairs of wild barley populations were calculated in genomic windows (size of 1 Mb, shift of 200 kb) using Hudson's estimator with the formula given as equation 10 (ref. 60) using a published Perl script[40]. Coverage-normalized SNP distances were calculated as described above in 1-Mb genomic windows (shift of 200 kb). Distributions of $\log_{10}$-transformed distances in the genomic compartments distal, interstitial and proximal were plotted for each wild barley population in R (v3.5.1)[61]. To infer divergence times, only SNPs in a 50-Mb region flanking the centromeres (±25 Mb) were used. SNP distances were converted into divergence times using the formula $g = d/2\mu$, where $g$ is the number of generations, μ is the mutation rate and $d$ is the number of SNPs per bp. We assumed that the generation time in the annual species $H.$ $vulgare$ is 1 year. We used a random mutation rate of $6.13 \times 10^{-9}$ as had been determined by Wang et al.[62] in the Pooideae grass $Brachypodium$ $distachyon$. The SNP number distribution was visualized by frequency polygons with logarithmic binning (number of bins of 50, range of $10^1$–$10^{4.5}$ (31,622 SNPs)).

## Demographic history of wild barley
Demographic inference was done with PSMC[8] (v0.6.5-r67, default parameters) using pseudo-diploid genomes, which were created by combining the BAM files of two homozygous individuals as previously described[63–65]. We performed two types of PSMC analyses. The first was conducted separately for five wild barley subpopulations to infer their respective demographic histories (Supplementary Fig. 2). The second treated all wild barley samples as a single population to capture the average demographic history of the species (Extended Data Fig. 3b). For the population-specific PSMC analysis, we first calculated the IBS distribution for all pairwise combinations of individuals within each group. On the basis of the distribution, IBS values were divided into two to four bins. Within each bin, we selected either all sample pairs (if the number of combinations was fewer than 50) or 50 pairs (if the number of combinations was more than 50) evenly distributed from low to high IBS values (Supplementary Table 22). In selecting sample pairs, we also considered the sequencing coverage of each individual. A pair was retained only if the ratio of coverage, defined as ratio = coverage$_{sample2}$/(coverage$_{sample1}$ + coverage$_{sample2}$), fell within the range 0.45–0.55. For the species-level PSMC analysis, the method was the same, except that each pair of samples was required to come from different subpopulations (Supplementary Table 23). PSMC is based on a panmictic model, assuming random mating, in which an individual (for example, a mammal) carries haplotypes derived from different ancestors. For selfing species, the outcome of pseudo-diploid PSMC is highly dependent on IBS. The higher the IBS, the closer the relationship between the pair, and the more likely the haplotypes come from a shared ancestor, which violates the assumption of random mating in PSMC. Conversely, pairs with lower IBS values are more likely to carry haplotypes from different ancestors, making them more consistent with the PSMC model. Therefore, we used the sample pairs from the lowest IBS bin (0.60 < IBS < 0.67) to represent the average demographic history of wild barley (Fig. 2b and Extended Data Fig. 3b).

## Analysis of deep divergence region on chromosome 5H
We used MUMmer (v4.0.0)[66] to align eight barley genome assemblies with different haplotypes[16] on chromosome 5H, 100–300 Mb. The minimum alignment identity was 90 and the minimum alignment length was 2,000 bp.

We used cross-population composite likelihood ratio (XP-CLR)[22], a method for detecting selective sweeps based on allele frequency differentiation, to assess whether a selective sweep signal exists in the deep divergence region on chromosome 5H. First, we performed genotype imputation and phasing of the SNP matrix using Beagle (v5.5)[67]. We then applied a Python implementation of XP-CLR (https://github.com/hardingnj/xpclr) to calculate XP-CLR scores between the southern Levant population and each of the other four wild barley groups. The analysis was performed using sliding windows of 1 Mb in size (shift of 200 kb). According to our previous definition (Extended Data Fig. 4b), excluding the three intermediate haplotypes, the remaining wild and domesticated barley samples were classified into two haplotype types: haplotype1 and haplotype2. Candidate genes were identified based on the SNP matrix constructed using the wild barley accession B1K-04-02 (FT11) as the reference genome[16] (Supplementary Fig. 4b). The effects of SNPs and indels residing in the genes of those regions were classified with SnpEff (v4.3t)[68], and variants with high allele frequency differentiation in haplotype1 and haplotype2 were prioritized (Supplementary Table 5).

### Definition of ancestral haplotype groups
AHGs were defined with IntroBlocker (v2)[9]. To determine an appropriate threshold for separating haplotypes, we computed coverage-normalized SNP-based distances in 1-Mb windows (shift of 200 kb): (1) among wild samples; (2) among domesticated samples; and (3) between wild and domesticated samples. In each of the three cases, all possible pairwise combinations of samples were considered. We selected a threshold of 400 SNPs per Mb to separate AHGs. Coverage normalized SNP–distance matrices computed from 367 high-coverage samples were used as input for IntroBlocker with the 'semi-supervised' model, giving precedence to wild over domesticated samples in the labelling of AHGs. IntroBlocker was run with different window sizes: 100 kb (shift of 20 kb), 500 kb (shift of 100 kb), 1 Mb (shift of 200 kb), 2 Mb (shift of 400 kb) and 5 Mb (shift of 5 Mb). The results of the 5-Mb run are shown in Fig. 3b and Extended Data Fig. 5b. After inspection of results (Supplementary Fig. 9), the results from the 100-kb (shift of 20 kb) run were used for downstream analyses.

### Analysis of the AHG matrix
The proportions of shared and private AHGs in wild and domesticated barleys were determined with custom Perl scripts. Saturation curves were calculated as follows. We chose sets of $k$ wild barleys (from a universe of 251 samples) at random, with $k$ ranging from 1 to 250. For each $k$, the selection was repeated 100 times. For each of the samples, we determined the proportion of haplotypes seen in the domesticate that were shared with that set. Mean values and 95% confidence intervals for each $k$ were calculated in R (v3.5.1)[61] based on the $t$-distribution (via the t.test() function). Two-dimensional haplotype frequency spectra were calculated with custom Perl scripts. Genomic windows with more than 20% missing data points were excluded.

To infer the times at which wild haplotypes entered the domesticated gene pool, we ran IntroBlocker with different thresholds for haplotype separation: 400 SNPs (equivalent to an approximate divergent time of 32,000 years ago), 98 SNPs (8,000 years), 73 (6,000 years), 49 SNPs (4,000 years) and 24 SNPs (2,000 years). For each domesticated haplotype, we compared the results from IntroBlocker runs with different thresholds (divergence time brackets). The latest bracket in which haplotype sharing between wild and domesticated samples occurred was considered a *terminus post quem* for when a wild haplotype type entered the domesticated gene pool. This method is agnostic about the direction of gene flow. To exclude recent introgressions from domesticated to wild barley, we removed windows in which multiple domesticated barley samples and a few wild barleys share haplotypes that diverged within the past 8,000 years. To determine the spatial origin of haplotypes, we averaged the ancestry ADMIXTURE coefficients of all wild individuals in which a given domesticated haplotype occurred

(Supplementary Fig. 29). If two wild samples that shared a domesticated haplotype were highly similar (pairwise IBS ≥ 0.95), only one was used for the calculation.

### Haplotype-based genetic diversity and selective sweeps
Saturation curves for the average number of haplotypes in a genomic window as a function of sample size were obtained by randomly selecting $k$ individuals with $k$ ranging from 1 to 115 for domesticated samples and from 1 to 250 for wild samples. For each $k$, the selection was repeated 100 times. Average haplotype numbers were determined for each subsample. Mean values and 95% confidence intervals were calculated in R (v3.5.1)[61] based on the $t$-distribution (via the t.test() function). $\theta_W$[59] and the Shannon diversity index[69] were calculated with a custom Perl script on haplotype matrices including only genomic windows with less than 20% missing data. The $\theta_W$ and Shannon index in seven barley chromosomes were plotted with Gnuplot using 'smooth bezier'.

We looked for regions of reduced diversity in domesticated relative to wild barley and therein searched for genes that might have been potential targets of selection. To not bias the analysis by the use of a domesticated reference genome (that of cultivar Morex), IntroBlocker was re-run using the SNP matrix based on the wild barley accession B1K-04-02 (FT11)[16]. Regions with a Shannon index ≤ 1 were considered selective sweeps. The effects of SNPs and indels residing in the genes of those regions were classified with SnpEff (v4.3t)[68], and variants with high allele frequency differentiation were prioritized.

The differentiation between populations of domesticated barley was assessed by computing the absolute allele frequency difference[70]. The following comparisons were done: NE + EU versus ETH, NE + EU versus Asia, ETH versus Asia, NE versus EUT, NE versus EUS, and EUT versus EUS. In addition, we calculated $F_{ST}$ in genomic windows (size of 100 kb, shift of 20 kb) using the same method as in wild barley. Allele frequency difference was used for haplotypes derived from high-coverage samples (SNP1); $F_{ST}$ calculations were performed for all samples, including low-coverage samples (SNP2).

### Demographic history of domesticated barley
Trajectories of effective population size across time were inferred with PSMC[8] (v0.6.5-r67, default parameters) using pseudo-diploid genome sequence from two homozygous barley individuals. A generation time of 1 year and a mutation rate of $6.13 \times 10^{-9}$ were used. We ran PSMC on 341 pseudo-haploid genomes obtained from all possible permutation of sample pairs from within 15 domesticated populations to reflect the population history of each subpopulation of domesticated barley. Given that domesticated barley originates from a mosaic genome composed of diverse wild barley lineages, we used the average demographic history of wild barleys (sample pairs from the lowest IBS bin between 0.60 and 0.67 in Extended Data Fig. 3b) as a reference background to compare temporal changes in effective population size ($Ne$) between 15 cultivated barley groups and wild barley.

Split times between pairs of domesticated barley populations were determined by inspecting the distributions of SNP numbers between pairs of samples in those windows (size of 1 Mb, shift of 200 kb) where a given pair of samples differed by fewer than 300 SNPs (corresponding to a divergence of 24,470 years). Only 1-Mb windows in which the intersection of effective covered regions between the two samples exceeds 0.9 Mb were retained for SNP number calculation.

The SNP number distribution was visualized by frequency polygons (linear binning; number of bins of 50; range of 0–300). SNP numbers were converted to divergence time using the following formula: time = (SNP number per Mb/$10^6$)/($2 \times 6.13 \times 10^{-9}$), where the $6.13 \times 10^{-9}$ was the random mutation rate ($\mu$) of *B. distachyon*[62].

### Validation of inferred split times
We used a previously published two-rowed ancient barley sample, JK3014[71] (approximately 6,000 years old, from Israel), to assess the

accuracy of our method (Supplementary Fig. 30). JK3014 was chosen because it is a high-depth sequenced sample (102×) and underwent uracil–DNA–glycosylase (UDG)[72] treatment, which reduces post-mortem DNA damage. JK3014 was jointly analysed with 116 high-depth modern barley samples for SNP calling. SNPs were filtered using the same pre-processing criteria that we applied in our SNP number calculation. We then calculated the SNP number between JK3014 and each of the 116 samples, without excluding C→T and G→A substitutions. The analysis used a 1-Mb sliding window (shift of 200 kb). To convert the SNP number to time, we used two models:

Model 1 assumes JK3014 is a direct ancestor of modern two-row Israel barley (ISR-THS). In this case, time = $d/\mu$, where $d$ equals the SNP number in 1-Mb windows/$10^6$ and $\mu$ is the mutation rate.

Model 2 assumes JK3014 and ISR-THS share a common ancestor, and their divergence time slightly predates 6,000 years ago. In this case, time = $d/(\text{coefficient} \times \mu)$. If JK3014 was a modern barley sample, the coefficient would be 2. Therefore, a reasonable estimate for this coefficient lies between 1 and 2. We used 1.2 to approximate a divergence time slightly earlier than 6,000 years ago. In addition, as UDG treatment cannot entirely eliminate ancient DNA damage, we assumed 10% of the C→T and G→A SNPs might be false positives. Thus, the final equation for model 2 becomes: time = $(d/1.1)/(1.2 \times \mu)$.

## Estimation of haplotype age for domestication genes

We used GEVA (v1)[30] to estimate the age of haplotypes associated with three domestication genes in barley. For GEVA, the alternative allele is assumed to be the derived allele. As the domesticated haplotypes of these genes in domesticated barley are all recessive mutations compared with wild barley, we used the SNP matrix based on the wild barley reference genome B1K-04-12 (FT11)[16]. This setup ensures that the causal variant of the domesticated haplotype is treated as the derived allele. Phasing of the SNP matrix was performed using Beagle (v5.5)[67]. For genes with known causal variants, we applied the following strategies to estimate haplotype age: if the causal variant is a SNP (for example, *vrs1.a3* and *ppd-H1*), we directly used GEVA to estimate the age of that SNP. If the causal variant is a short indel (for example, *btr1*, *btr2*, *vrs1.a1* and *vrs1.a2*), we constructed pseudo-SNPs at the indel position (for example, for the 1-bp deletion at position 41,130,358 in *btr1*, C/−), such as C→A, C→T and C→G, and estimated their ages using GEVA.

In both SNP and indel cases, we also identified haplotype-specific private SNPs that are in complete linkage with the causal variant and used these SNPs to estimate haplotype age. The defining feature of a causal variant is that it is private to the focal population and has a genotype frequency of 100%. 'Private' refers to those found exclusively in the focal haplotype relative to all other barley samples, including both wild and domesticated barley. The SNPs that we selected as being in 'complete linkage with the causal variant' share these same characteristics: they are private to the population and occur at a genotype frequency of 100%. Therefore, these SNPs probably originated either before or concurrently with the causal variant and can be used alongside it to estimate the age of the haplotype. The actual age of the haplotype is thus equal to or later than the age estimated by this method. For each haplotype, we randomly selected approximately 40 private SNPs, as well as the causal SNP or pseudo-causal SNPs for the calculation (Supplementary Table 14). For large deletions (for example, *Nud*), haplotypes with unknown causal variants (for example, *vrs1.a4*) and functional (dominant) haplotypes in cultivated barley (*Vrs1.b2*, *Vrs1.b3* and *Nud*), we estimated haplotype age using approximately 40 private SNPs specific to the domesticated haplotypes. To avoid confounding effects from recombination, we excluded all domesticated samples showing evidence of recombinant haplotypes in the regions of interest (Supplementary Table 13). GEVA analyses were performed using default parameters, and downstream filtering was conducted using the 'estimate.R' script provided in the GEVA package. The mutation rate that we used is $6.13 \times 10^{-9}$ from *B. distachyon*[62]. For each SNP, ten replicate runs were performed with different random seeds. Because recombinant haplotypes were excluded from the domesticated haplotype analyses, we reported haplotype ages based on the mutation clock model. Finally, given that barley is a highly selfing species with negligible heterozygosity (that is, nearly haploid in effect), and GEVA was originally developed under a diploid model (for human data), we multiplied all age estimates by 2 to account for ploidy differences and to report the final haplotype age.

As a control group, for each gene locus, we randomly selected approximately 40 SNPs (0.2 < allele frequency < 0.5) from wild barley within the same genomic region and estimated their ages (Supplementary Table 15). Given their uncertain origin − either recent or ancient in the absence of selection − low-frequency SNPs are less suitable as reliable controls. By contrast, high-frequency SNPs (for example, those with frequencies above 20%) are likely to have arisen in the past and become fixed or nearly fixed in the population, and thus are expected to exhibit older ages. For wild barley SNP, the joint mutation and recombination clock model were used. In addition, the recessive *ppd-H1* allele, which may predate domestication[34], was also included as a control group.

To infer the most likely spatial origins of three genes, a neighbour-joining tree for each gene was constructed with SNPs from an interval within their sweep region. For the *btr1/2*, *vrs1* and *nud* loci, the interval extended from 39.4 to 39.7 Mb on chromosome 3H, from 570.5 to 517.2 Mb on chromosome 2H and 525.3−525.7 Mb on chromosome 7H, respectively. The neighbour-joining tree was constructed using SNPs based on the MorexV3 reference (SNP1).

## Archaeological excavations

We analysed ancient DNA sequences of 23 barley grains excavated at three archaeological sites in Israel (Supplementary Table 16). This number included published data of five barley grains from Yoram Cave[71]. Archaeobotanical procedures were performed as described by Lev-Marom et al. (manuscript in preparation). The sites Yoram Cave and Timna 34 have been described by Mascher et al.[71] and Lev-Marom et al. Abi'or Cave is a medium-sized cave located on the eastern slopes of the Judean Desert, above Jericho, approximately 50 m below sea level, across from the Karantal Monastery. The excavations at the cave were directed by the late H. Eshel in 1986. It is situated above a larger cave known as 'The Spies Cave' and has three openings above it. The cave contains a main long tunnel, approximately 50 m long, and has revealed archaeological material dating from the Chalcolithic period to the time of the Bar Kochba Revolt (2nd century CE). The cave was found to be heavily disturbed by animals, antiquities robbers and monks who lived in it during the Islamic and more recent periods.

## Ancient DNA sequencing and analysis

All laboratory procedures for sampling, DNA extraction, library preparation and library indexing were conducted in facilities dedicated to ancient DNA work at the University of Tübingen. Before DNA extraction, all seeds were cut into two parts: one part of each seed (36-6.5 mg) was used for DNA extraction and further processing, the other part (26-3.4 mg) was used for radiocarbon dating at the Klaus-Tschira-Archäometrie-Zentrum, Curt-Engelhorn-Zentrum Archäometrie gGmbH. DNA extraction was then performed according to a well-established extraction protocol for ancient plant material[71] and double-stranded dual-indexed DNA libraries were produced[73,74]. Six ancient DNA samples (TU697 and JK2281-JK3014) were treated with UDG[72] before sequencing. Sequencing was done on Illumina devices at IPK Gatersleben, the University of Tübingen and the Max-Planck Institute or the Science of Human History Jena.

Paired-end Illumina reads of each sample were merged with leeHom (v1.2.17)[75] and mapped to the MorexV3 genome sequence assembly[15] using Minimap2 (v2.24)[48]. BAM files were sorted and duplicates were marked with Novosort (v3.06.05; https://www.novocraft.com/products/novosort/). Nucleotide misincorporation profiles were generated

with mapDamage (v2.0.8)[76]. Variant calling was done with bcftools (v1.15.1)[49] using the command 'mpileup -a DP,AD -q 20 -Q 20 --ns 3332'. We omitted the parameter '--variants-only' in 'bcftools call' to output genotype in all sites. C→T and G→A were excluded, where the C and G are the alleles in the reference genomes and T and A are the alternative alleles called from the short-read data. The resultant SNP matrix was merged with the three different SNP matrices: SNP1 (367 high-coverage samples), SNP2 (302 domesticated barleys) and a published SNP matrix constructed from GBS data of 19,778 domesticated barleys[24]. The GBS matrices had been filtered for site-level missing rate (less than 20%) before merging. The merged SNP1 matrix was used for PCA with smart-PCA (v7.2.1)[44] using the parameter 'lsqproject: YES'. Neighbour-joining trees were constructed using only SNPs in a 50-Mb region flanking the centromeres (±25 Mb) on each of the seven chromosomes and including only six high-coverage ancient DNA samples, to determine the proximal haplotypes of ancient barley. The merged GBS matrix was used to compute an IBS matrix with PLINK (v1.9)[45]. To examine the phylogenetic relationships between ancient DNA and modern domesticated barley, we constructed genome-wide phylogenetic trees using two merged SNP datasets: SNP1 and SNP2, each incorporating ancient DNA samples.

To compare genetic diversity between individual ancient and modern barley samples without relying on population-level statistics, we leveraged rare alleles identified in a comprehensive wild barley panel as proxies for ancestral diversity (Supplementary Fig. 18). Wild barley has the most extensive reservoir of allelic variation; alleles with very low frequency in this panel (for example, $0 < MAF \le 0.01$) are unlikely to persist through strong bottlenecks or selective sweeps, and thus serve as sensitive markers of lost diversity. For each sample pair, we counted the number of these wild-derived rare alleles present in the ancient genome (A) and in the modern genome (M), and defined the 'relative diversity change' as $(M − A)/A$. A positive value indicates retention or gain of ancestral diversity in the modern sample, whereas a negative value signifies diversity loss relative to the ancient sample. This approach allows us to quantify diversity change at the single-sample level in a straightforwards, interpretable manner, without requiring large cohort sizes or population-based diversity estimators. We calculated the relative change in genetic diversity between six high-coverage ancient samples and modern domesticated barley individuals from 15 populations.

The merged SNP1 and SNP2 were also used for the calculation of $D$ statistics with the qpDstat program of ADMIXTOOLS (v3.0)[77]. On the basis of previous phylogenetic analyses, we identified ISR-THS as the closest modern barley population to both Yoram Cave and Timna 34, and ME-SHS as the closest to Abi'or Cave. To test for potential gene flow between ancient and modern barley, we performed the following three $D$ statistics analyses: $D$ (ISR-THS, Yoram Cave; P3, *H. pubiflorum*), $D$ (ISR-THS, Timna 34; P3, *H. pubiflorum*) and $D$ (ME-SHS, Abi'or Cave; P3, *H. pubiflorum*). Here P3 refers to any of the 14 modern barley populations other than ISR-THS or ME-SHS, and *H. pubiflorum* is the outgroup.

### Reporting summary
Further information on research design is available in the Nature Portfolio Reporting Summary linked to this article.

## Data availability
The sequence data collected in this study have been deposited at the European Nucleotide Archive (ENA)[78] under BioProjects PRJEB65046, PRJEB56087 and PRJEB53924. The SNP and indel variant matrix will be available at the European Variation Archive[79] under BioProject PRJEB79752. ENA accession codes for individual genotypes are listed in Supplementary Table 1. AHG matrices have been deposited in the Plant Genomics and Phenomics Research Data Repository[80] (https://doi.org/10.5447/ipk/2025/7).

## Code availability
The shell and Perl scripts used in this study are available on GitHub (https://github.com/guoyu-meng/barley-haplotype-script).

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

**Acknowledgements** We are grateful for the technical assistance of S. König and I. Walde; the IT support by T. Münch and J. Bauernfeind; Z. Wang and W. Guo for their advice on the use of IntroBlocker; X. Liu and M. Schreiber for sharing knowledge on the archaeology of barley domestication and dispersal; and A. Börner who supported the development and maintenance of diversity panels. N.S. and M.M. were supported by grants from the German Ministry of Research and Education (BMBF; 031B0190 and 031B0884). B.J.S. is supported by the Lieberman-Okinow Endowment at the University of Minnesota. V.J.S. was supported by the University of Zurich's University Research Priority Program 'Evolution in Action: From Genomes to Ecosystems'. E.W. is supported by the Israel Science Foundation (grants 1179/13 and 545/23).

**Author contributions** M.M., Y.G. and N.S. designed the study. A.H. and N.S. conducted and supervised the sequencing. E.W. performed and supervised the archaeobotanical work. E.B.-Y. and U.D. performed the archaeological excavations. M.D., M.K., A.H.-S. and E.W. performed the archaeobotanical analyses. V.J.S., E.R. and J.K. performed and supervised the ancient DNA experiments. T.F. and B.J.S. contributed the sequencing data. Y.G. and M.J. analysed the data. Y.G. and M.M. wrote the manuscript. All authors edited the manuscript.

**Funding** Open access funding provided by Leibniz-Institut für Pflanzengenetik und Kulturpflanzenforschung (IPK).

**Competing interests** The authors declare no competing interests.

**Additional information**
**Correspondence and requests for materials** should be addressed to Ehud Weiss or Martin Mascher.

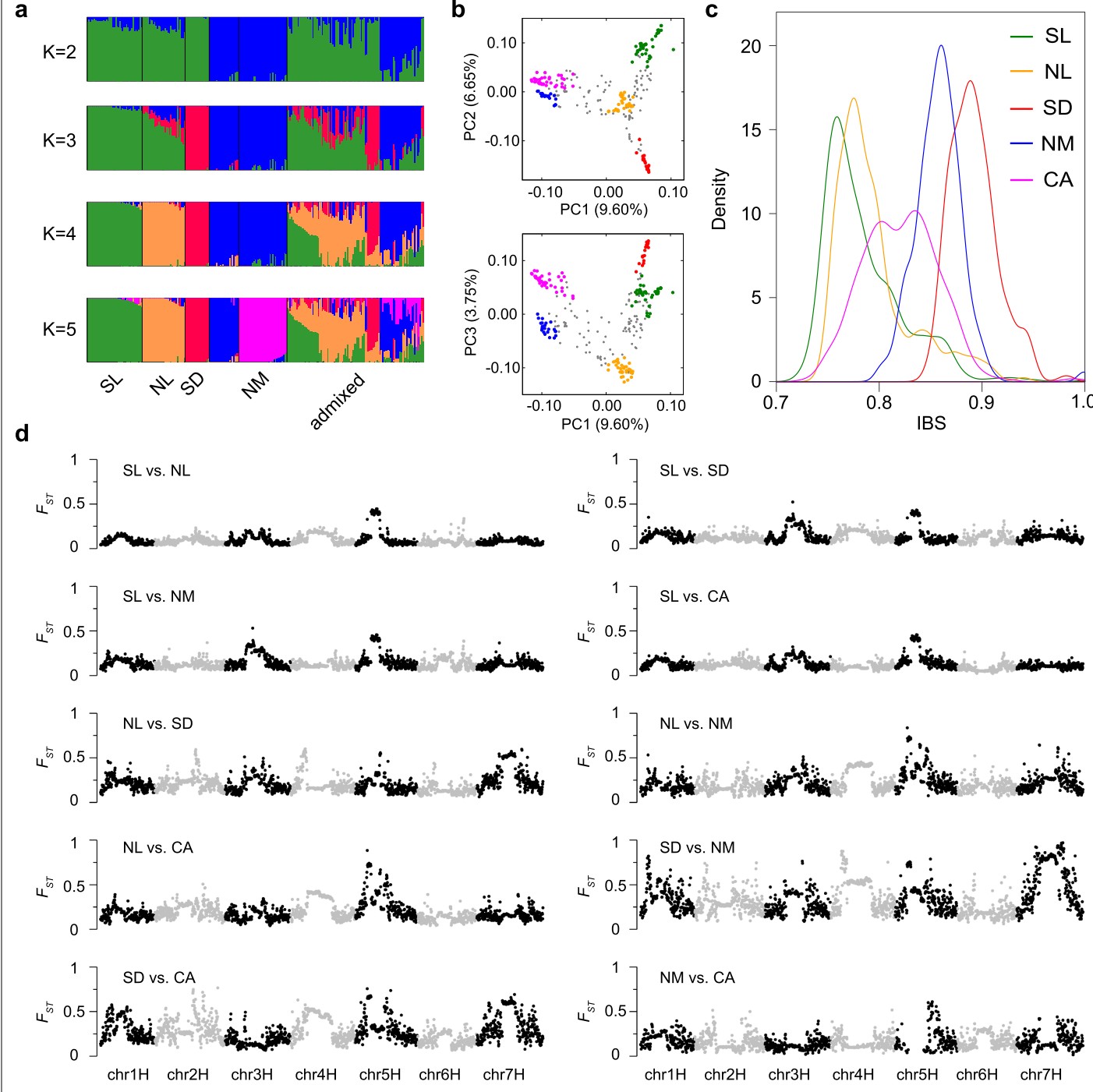

**Extended Data Fig. 1 | Population structure of wild barley.** (**a**) Individual ancestry coefficients in ADMIXTURE with the number of ancestral populations (*K*) ranging from 2 to 5. Individuals with ≥ 85% ancestry form were considered un-admixed samples. (**b**) Principal component analysis (PCA) based on 37.14 million biallelic SNPs with a MAF > 5%. The first three PCs are shown (PC1 vs. PC2 and PC1 vs. PC3). Un-admixed samples are colored to their major ancestry component in panel (**a**). (**c**) Distribution of pairwise identity-by-state (IBS) in the five wild barley populations. (**d**) Fixation indices (*F*$_{ST}$) in 1 Mb windows (shift: 200 kb) along the genome. All possible contrasts between any two of the five wild barley populations are shown.

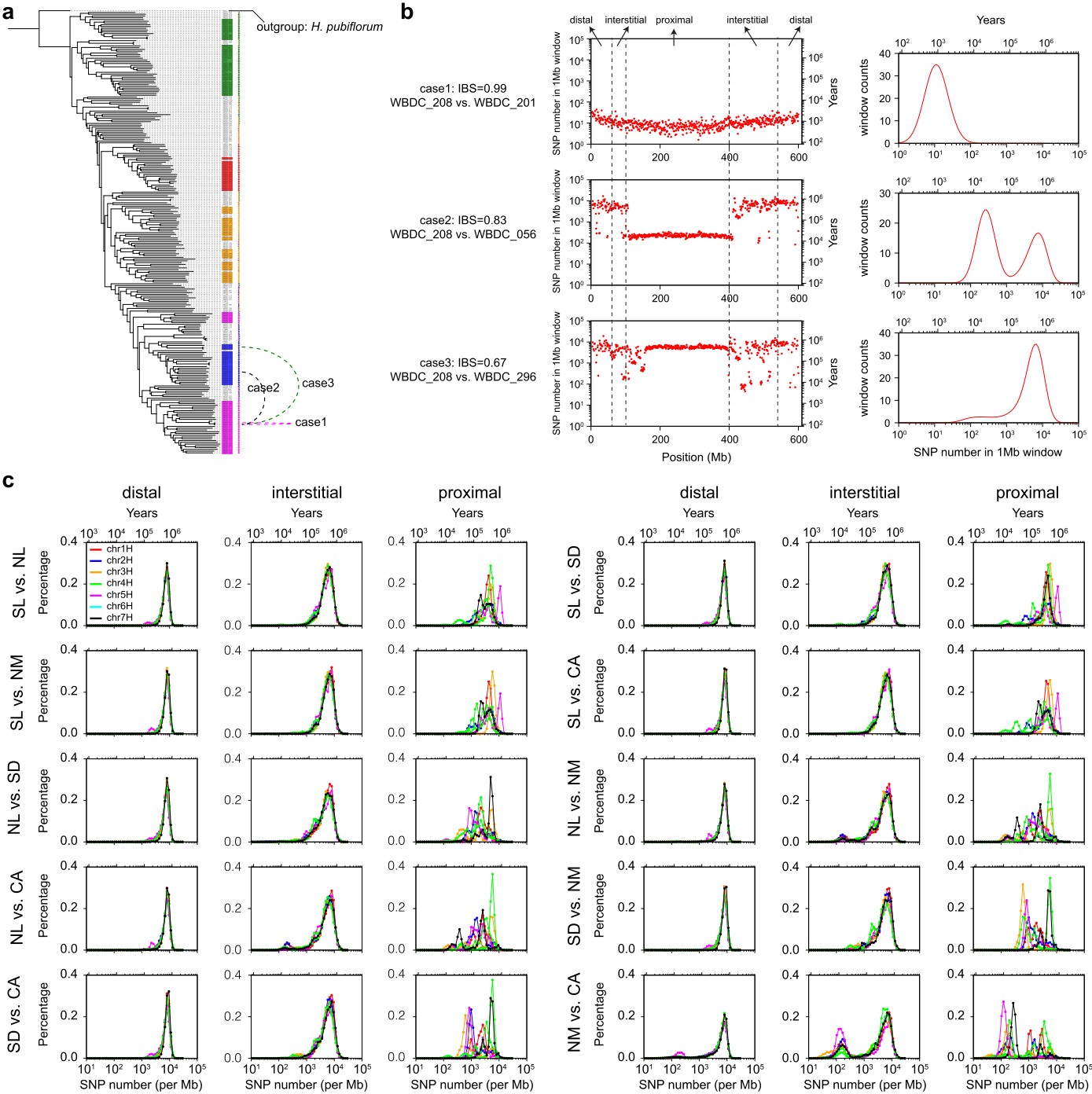

**Extended Data Fig. 2 | Sequence divergence between wild barleys in different genomic compartments.** (**a**) Neighbor-joining tree of 251 wild barley genotypes computed from 37.14 M biallelic SNP markers with a MAF > 5%. *H. pubiflorum* was used an outgroup. Pie charts to the right show ancestry coefficients as determined by ADMIXTURE (Extended Data Fig. 1). The colored sample labels indicate the populations to which unadmixed samples belong. (**b**) The left-hand panels show the sequence divergence (SNPs per non-overlapping 1 Mb genomic windows) on chromosome 4H between three pairs of wild barleys. Case 1 each compares two members from the same population (CA, WBDC208 vs. WBDC201). In cases 2 and 3, WBDC208 is compared to two members of the NM population, WBDC296 and WBDC056. The right-hand panels show the respective distribution densities. (**c**) Distributions of sequence divergence (SNPs per Mb) in three genomic compartments in pairwise comparisons between members of different wild barley populations.

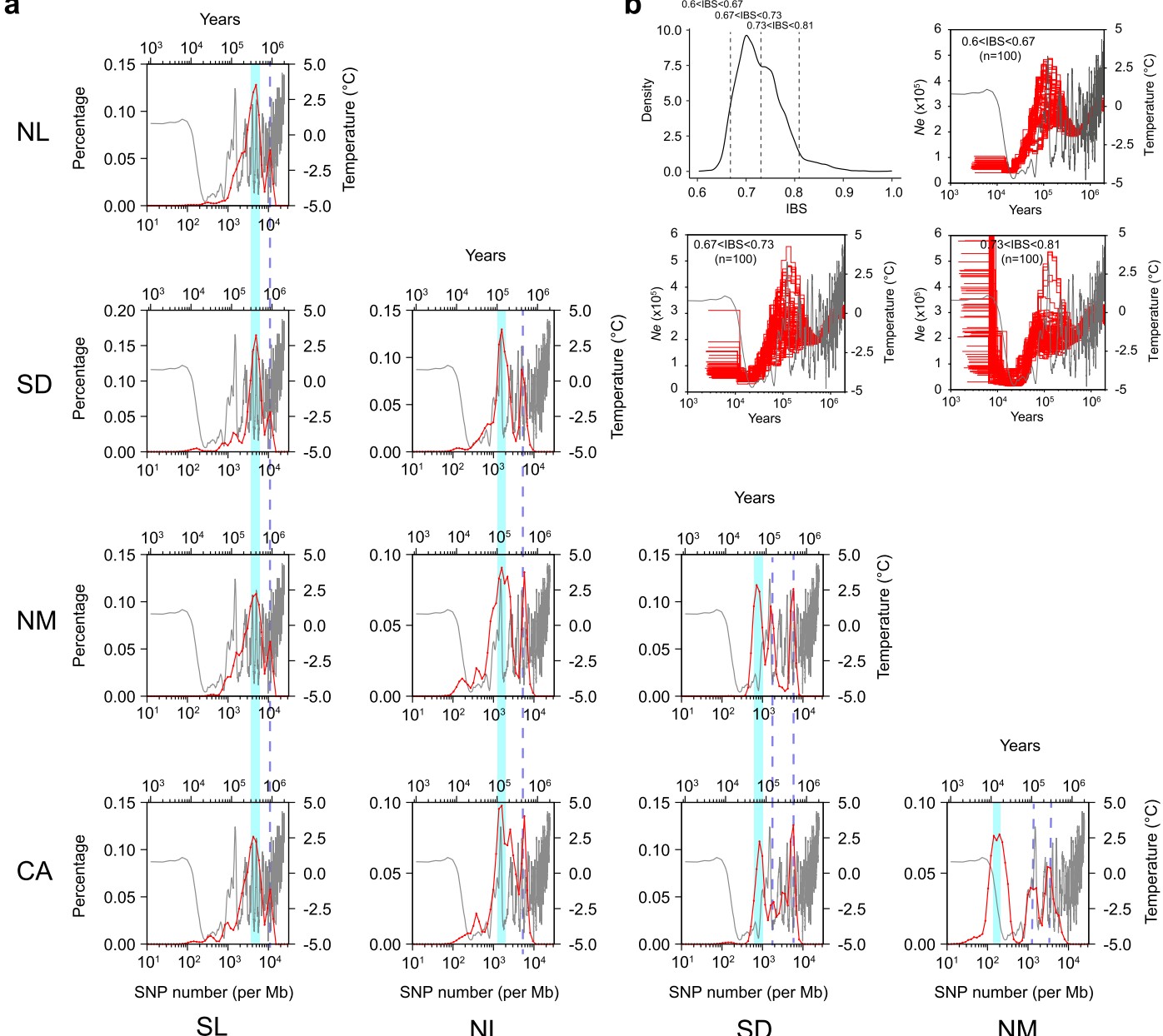

**Extended Data Fig. 3 | Demographic history of wild barley populations.**
(**a**) Distributions of pairwise sequence divergence (SNPs Mb) in pericentromeric regions (centromere +/− 25 Mb). The distributions were computed from comparisons between all possible combinations between members of the respective populations. Cyan shading marks the most recent peaks in the distributions interpreted as the most recent divergence from common ancestors. Dashed lines mark earlier such events. (**b**) Historic trajectories of effective population sizes computed with PSMC considering all wild barley as a population. The top-left panel shows the density distribution of pairwise Identity-By-State (IBS) values among all wild barley samples. We divided the IBS values into three bins. The remaining three panels display the PSMC results based on 100 randomly selected sample pairs from each bin, used to construct pseudo-diploid genomes. The y-axis represents the effective population size (Ne), and the x-axis represents time. Gray lines in panels (**a**) and (**b**) show the global average surface temperatures[39].

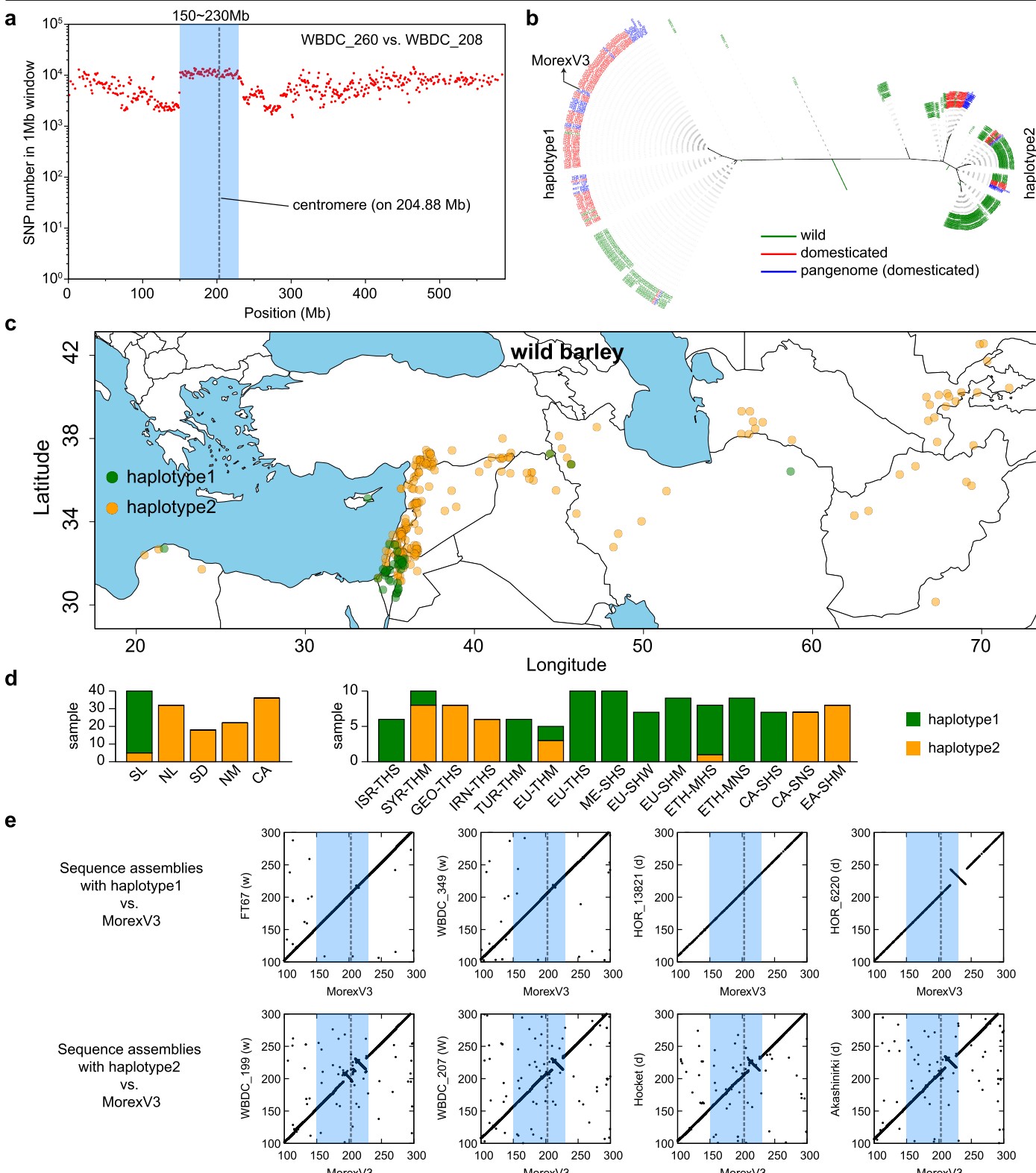

**Extended Data Fig. 4 | Deeply diverged pericentromic haplotypes on chromosome 5H.** (**a**) Sequence divergence (SNPs per Mb) on chromosome 5H between wild barleys WBDC260 (SL) and WBDC208 (CA). (**b**) Neighbor-joining tree of wild and domesticated barleys including 48 samples from the pangenome of Jayakodi et al.[16] based on the SNPs between 150–230 Mb on chr5H. (**c**) Collection sites of wild barleys carrying either haplotype. The "green" haplotype is most common in the SL population. Geographical outlines were obtained from the R package 'maps' (https://CRAN.R-project.org/package=maps), which uses public domain base map data (under a GNU General public license: version 2). (**d**) Frequencies of both haplotypes in wild and domesticated barley populations. (**e**) Alignments on chromosome 5H, 100 to 300 Mb between sequence assemblies of wild (w) and domesticated (d) pangenome accession with either haplotype to the MorexV3 reference. The accession names are indicated on the y-axis. In panels (**a**) and (**e**), the boundaries of divergent haplotypes (150 to 230 Mb) are marked by blue shading and the dashed line at 205 Mb indicates the position of the centromere in the MorexV3 reference.

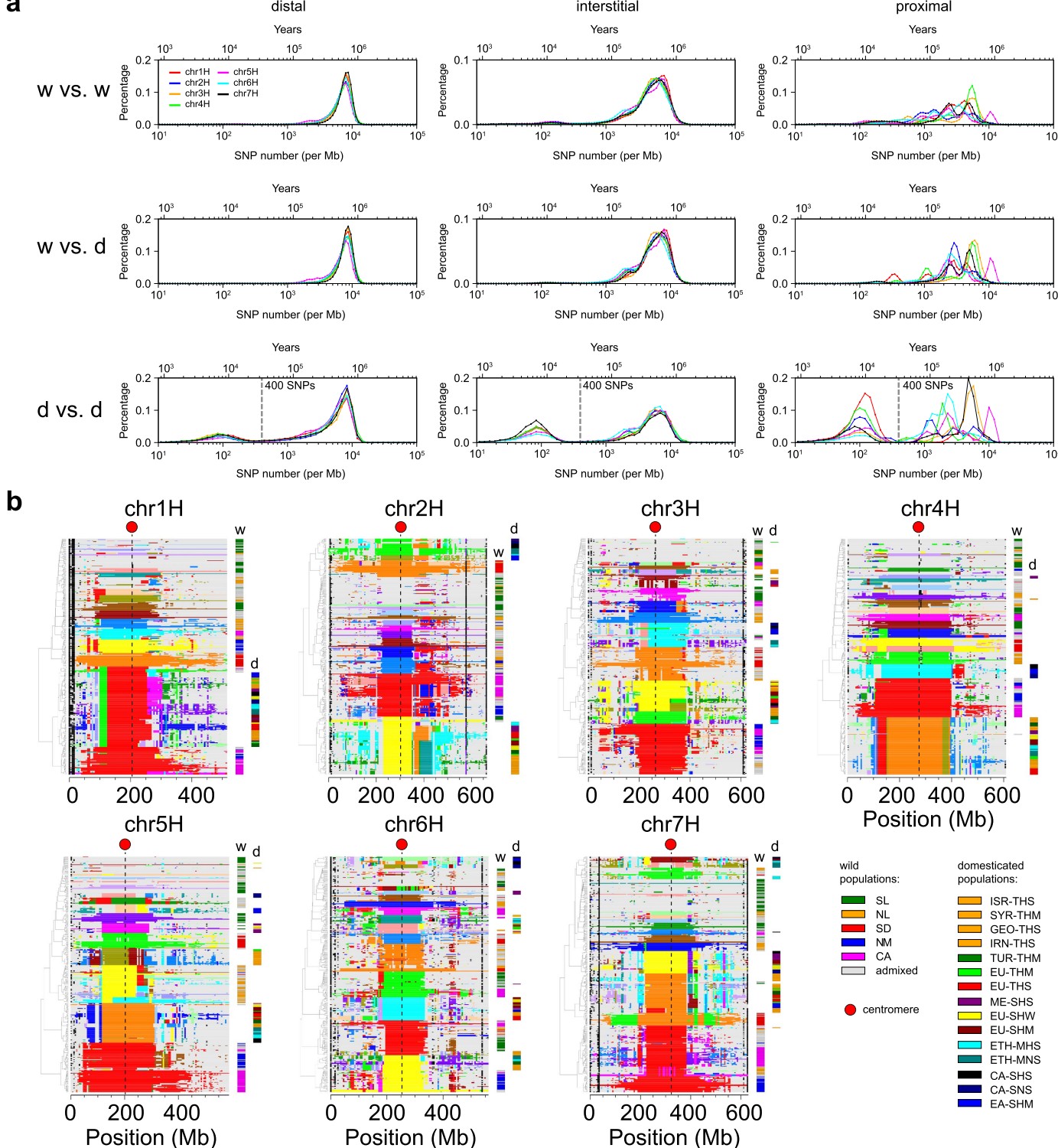

**Extended Data Fig. 5 | Ancestral haplotype groups (AHGs) in domesticated barley.** (**a**) Finding a threshold for defining AHG. Sequence divergence (SNPs per Mb) in pairwise comparisons between wild (w) and (d) domesticated barleys. A value of 400 SNPs per Mb was chosen as the threshold to separate haplotypes in IntroBlocker. (**b**) Mosaic view of AHGs on the seven chromosomes of barley. The data shown are from an IntroBlocker run with a 5 Mb window size (shift: 5 Mb). Colors were assigned to 20 most frequent AHGs by IntroBlocker in semi-supervised mode giving priority to wild over domesticated samples. Black color indicates missing data, gray stands for less frequent haplotypes. Colored bars on the right-hand side of each sub-panel assign samples to wild (w) or domesticated (d) subpopulations according to the legend at the bottom right.

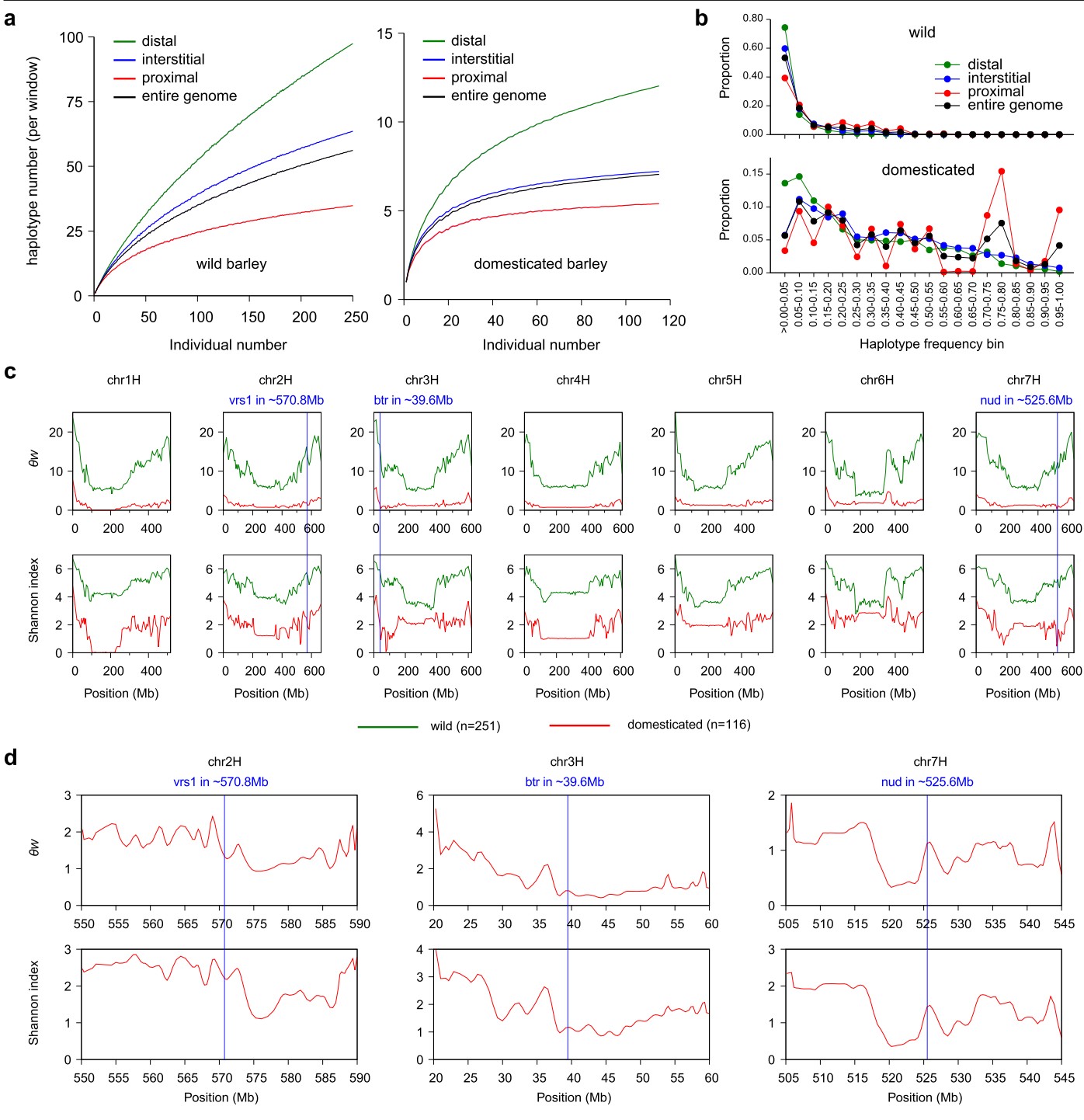

**Extended Data Fig. 6 | Haplotype-based diversity statistics. (a)** Number of observed distinct haplotypes per genomic window in wild and domesticated barley as a function of sample size. The solid line and shaded area represent, respectively, the average and 95% confidence interval of 100 random sub-samples. **(b)** Haplotype frequency spectra (bin size: 0.05) in wild and domesticated barley in different genomic compartments. **(c)** Watterson's θ and the Shannon index along the seven chromosomes of barley in wild and domesticated forms. Blue lines mark the location of *Btr1/2*, *Vrs1* and *Nud* loci. **(d)** Haplotype diversity ($θ_w$ and Shannon index) in domesticated barley around these loci.

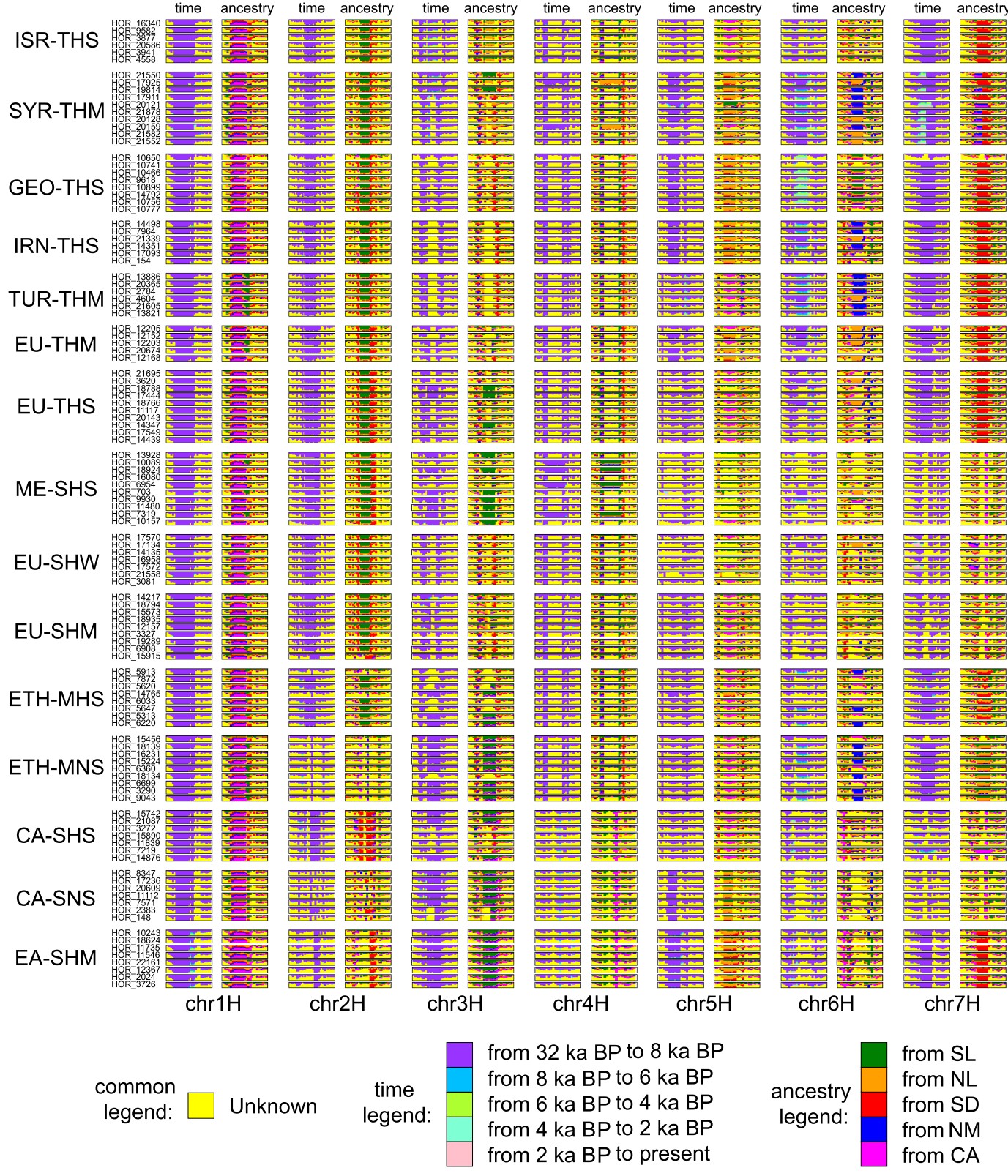

**Extended Data Fig. 7 | Spatiotemporal origins of haplotypes in domesticated barley.** The inferred times at which haplotypes entered the domesticated gene pool and the most likely wild source populations are shown along the genome (20 Mb windows) for 116 domesticated barleys from 15 populations. Colors correspond to periods (time) and population (ancestry) as indicated in the legend. Yellow color indicates unknown origins.

common legend: Unknown

time legend:
from 32 ka BP to 8 ka BP
from 8 ka BP to 6 ka BP
from 6 ka BP to 4 ka BP
from 4 ka BP to 2 ka BP
from 2 ka BP to present

ancestry legend:
from SL
from NL
from SD
from NM
from CA

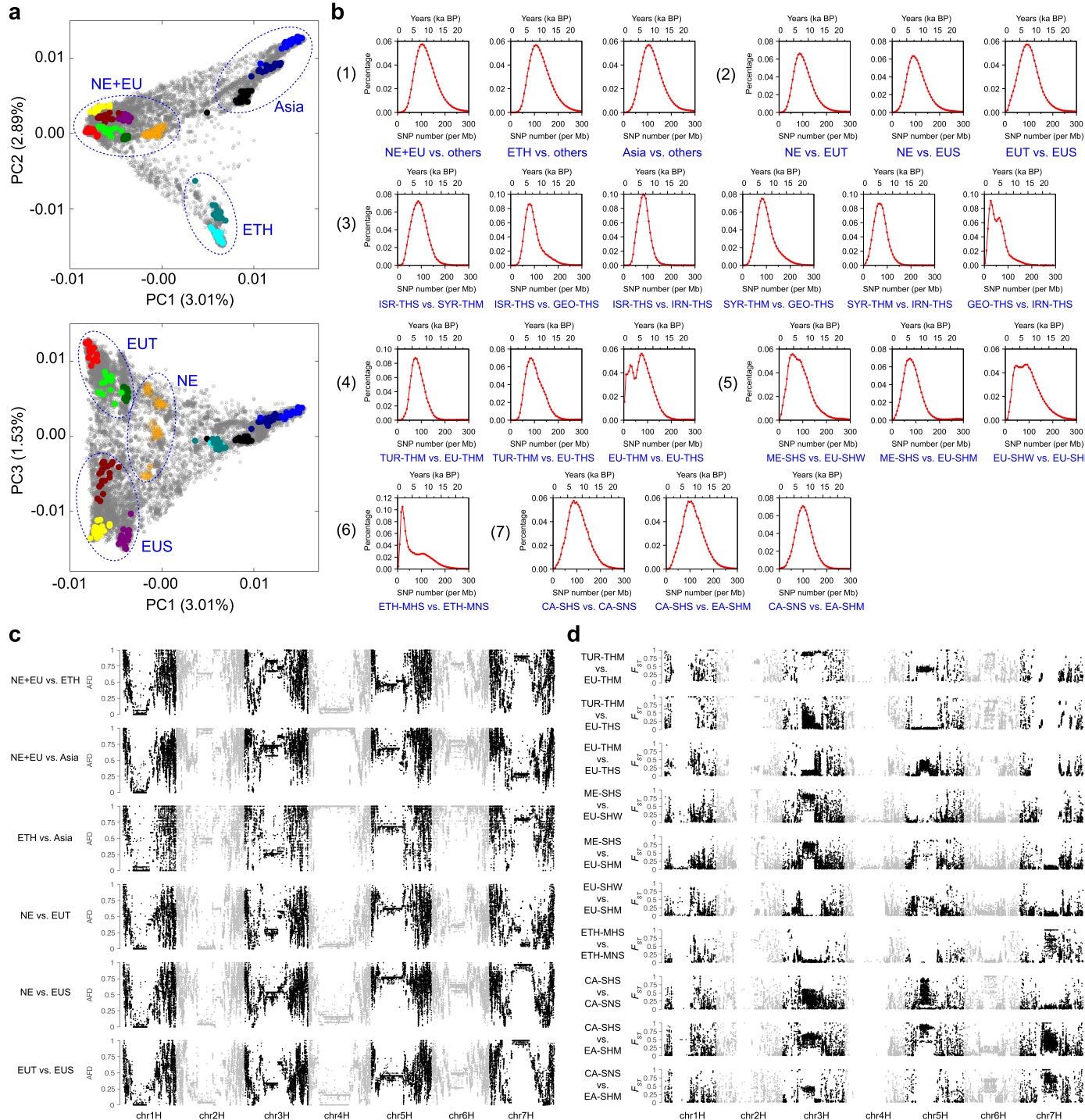

**Extended Data Fig. 8 | Divergence of domesticated barley populations.**
(**a**) Principal component analysis of 19,778 based on genotyping-by-sequencing data of Milner et al.[24] (62,888 biallelic SNPs). Samples analyzed in this study are shown in non-gray color. Blue circles delineate the groups used for the comparisons in panel (**b**): NE – Near East; EU – Europe and Mediterranean Basin; ETH – Ethiopia; Asia – Central and East Asia; EUT – EU two-rowed; EUS – EU six-rowed. (**b**) Distribution of sequence divergence between populations of domesticated barley. The comparisons are indicated in blue font below the sub-panels. (**c**) Absolute allele frequency difference (AFD) between different domesticated barley populations in sliding windows (size: 100 kb, shift: 20 kb) along the genome. AFD was computed on the haplotype matrix of high-coverage (~10x) samples. (**d**) $F_{ST}$ in sliding windows (size: 100 kb, shift: 20 kb) along the genome. $F_{ST}$ was computed from the SNP matrix of all samples (matrix SNP2, see Methods).

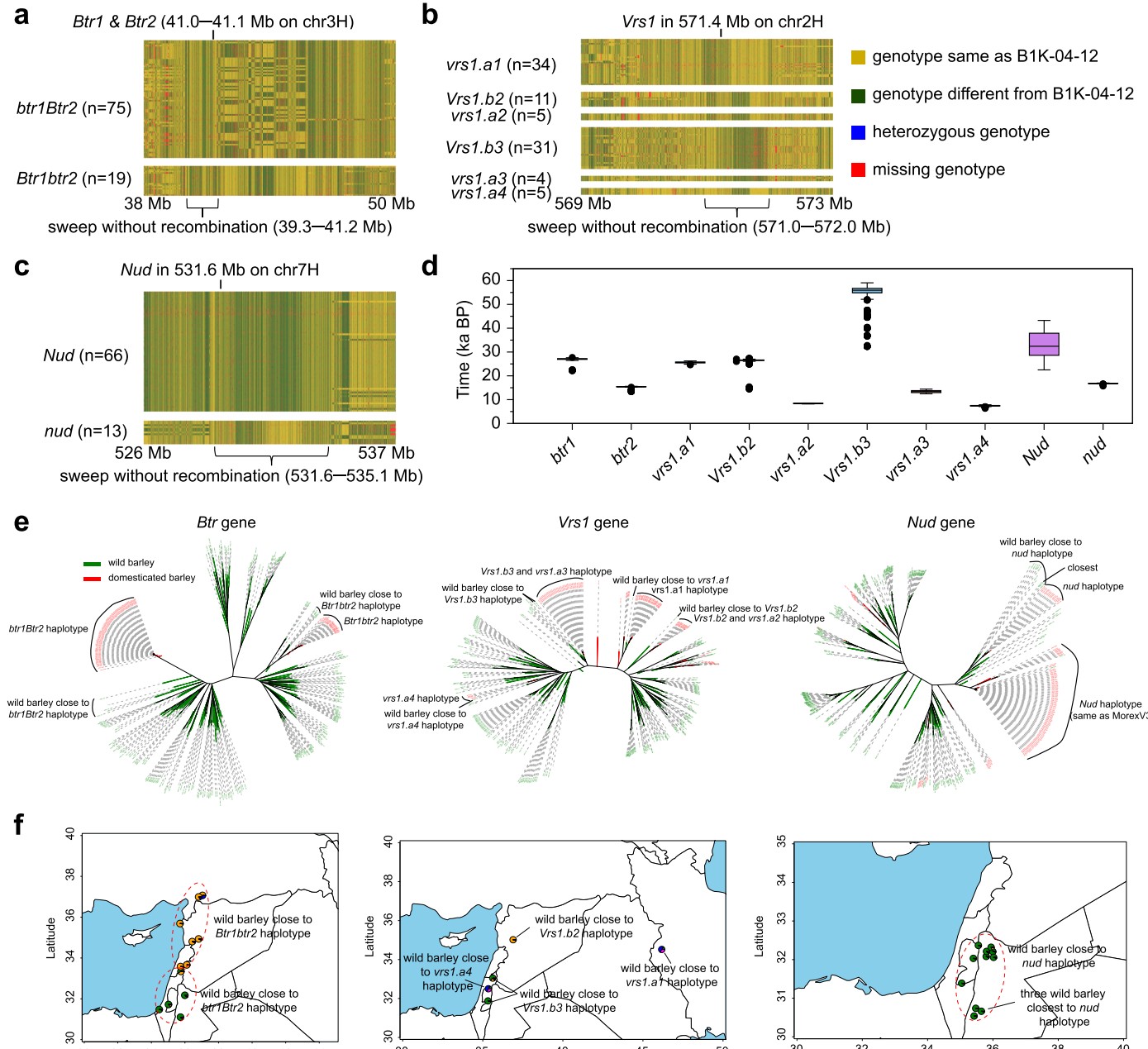

**Extended Data Fig. 9 | Origins of haplotypes of domestication genes.**
(**a**–**c**) SNP haplotypes surrounding the *Btr1/Btr2*, *Vrs1*, and *Nud* loci, respectively. Only domesticated barley accessions lacking recombination events within the sweep regions were included. (**d**) Box plots showing the estimated ages of causal and private SNPs within each mutant haplotype, as inferred by GEVA. Each box plot is based on n = S × 10, where S represents the number of SNP sites used, and the data were generated by repeating the analysis 10 times with different random seeds. Specifically, for each allele, the number of data points used to generate the box plots is as follows: *btr1* (n = 40 × 10), *btr2* (n = 42 × 10), *vrs1.a1* (n = 43 × 10), *Vrs1.b2* (n = 39 × 10), *vrs1.a2* (n = 3 × 10), *Vrs1.b3* (n = 41 × 10), *vrs1.a3* (n = 1 × 10), *vrs1.a4* (n = 40 × 10), *Nud* (n = 39 × 10), and *nud* (n = 40 × 10).

Box plots show the median (center line), the 25th and 75th percentiles (box bounds), and whiskers extend to values within 1.5× the interquartile range (IQR); outliers beyond this range are shown as individual points. (**e**) Neighbor-joining tree constructed from SNPs within the sweep intervals shown in panels (**a**–**c**), including both wild and domesticated barley accessions. (**f**) Geographic collection sites of wild barley samples whose haplotypes at the three loci are most closely related to those of domesticated barley. Geographical outlines were obtained from the R package 'maps' (https://CRAN.R-project.org/package=maps), which uses public domain base map data (under a GNU General public license: version 2).

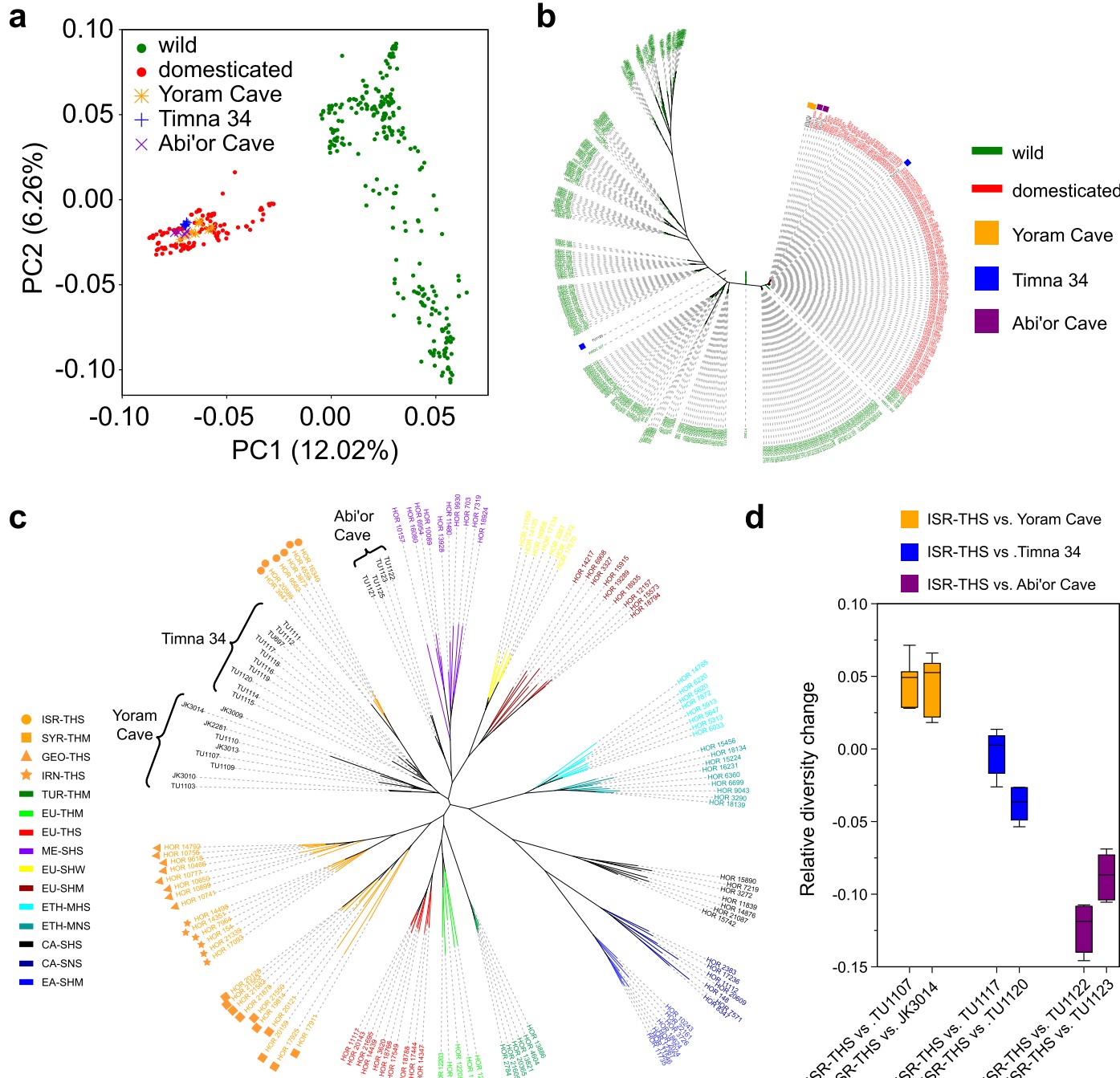

**Extended Data Fig. 10 | Ancient samples in the diversity space of extant barleys.** (**a**) PCA on high-coverage wild and domesticated samples onto which 23 ancient barleys were projected. (**b**) Phylogenetic tree illustrating haplotypes around the pericentromeric region on chr1H. High-coverage sequencing data were used from wild barley (n = 251), domesticated barley (n = 116), and ancient barley (n = 6). SNPs located within ±25 Mb of the centromere were used for tree construction. (**c**) Unrooted neighbor-joining showing the relationships between 23 ancient and 116 high coverage modern domesticated barley accessions. (**d**) Relative diversity change between modern Israel two-rowed barley (ISR-THS) and ancient barley accessions from three Israel sites. Each box plot is based on n = 6 biologically independent sample pairs, each consisting of one modern and one ancient barley accession. A positive value indicates an increase in diversity in modern barley relative to ancient barley, while a negative value indicates a decrease. Box plots show the median (center line), the 25th and 75th percentiles (box bounds), and whiskers extend to values within 1.5× the interquartile range (IQR); outliers beyond this range are shown as individual points.

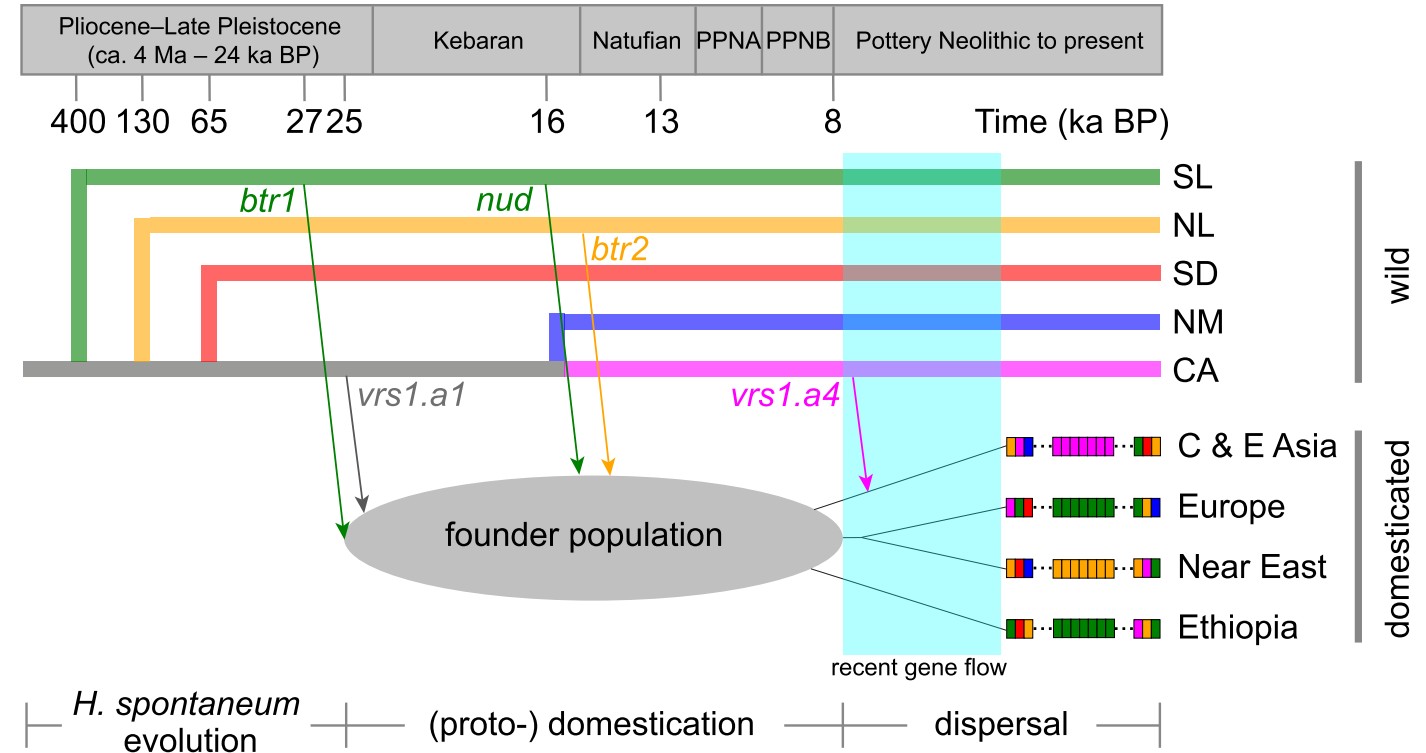

**Extended Data Fig. 11 | Schematic diagram of barley crop evolution.** The five colors in the tree on top represent the five wild barley populations (SL, NL, SD, NM and CA as shown in Fig. 1a), and their split times are based on results from Fig. 2e. Arrows correspond to domestication-related genes. The starting point of each arrow marks the estimated time when the domestication mutation emerged (Extended Data Fig. 9d) and from which wild barley population it originated. Note that the tip of the arrow of each arrow is symbolic—it does not represent the exact timing of entry into the founder population. Gene flow— facilitated by human migration or cultural exchange—gave rise to an admixed founder population. As populations of human farmers and their crops expanded outward from the Fertile Crescent, this founder population split into several geographically isolated lineages, each retaining chromosomal segments with diverse wild ancestry. The divergence of domesticated barley from this founder population is referenced from Fig. 5b. The rectangles in the bottom-right part represent chromosomes of four domesticated barley clusters (Near East, Europe, Ethiopia and C & E Asia) divided into windows, with dashed lines separating distal and proximal regions. C & E Asia refers to Central and Eastern Asia domesticated barley population. The cyan-colored rectangles indicate recent (<8 ka BP) gene flow from wild to domesticated barley in regions of sympatry. The timeline at the top shows the chronological sequence of agricultural cultures in the Near East and was adapted from Zeder et al.[81].

# Reporting Summary

## Statistics

For all statistical analyses, confirm that the following items are present in the figure legend, table legend, main text, or Methods section.

| n/a | Confirmed | |
|---|---|---|
| ☐ | ☒ | The exact sample size (*n*) for each experimental group/condition, given as a discrete number and unit of measurement |
| ☐ | ☒ | A statement on whether measurements were taken from distinct samples or whether the same sample was measured repeatedly |
| ☐ | ☒ | The statistical test(s) used AND whether they are one- or two-sided *Only common tests should be described solely by name; describe more complex techniques in the Methods section.* |
| ☒ | ☐ | A description of all covariates tested |
| ☒ | ☐ | A description of any assumptions or corrections, such as tests of normality and adjustment for multiple comparisons |
| ☐ | ☒ | A full description of the statistical parameters including central tendency (e.g. means) or other basic estimates (e.g. regression coefficient) AND variation (e.g. standard deviation) or associated estimates of uncertainty (e.g. confidence intervals) |
| ☐ | ☒ | For null hypothesis testing, the test statistic (e.g. *F*, *t*, *r*) with confidence intervals, effect sizes, degrees of freedom and *P* value noted *Give P values as exact values whenever suitable.* |
| ☒ | ☐ | For Bayesian analysis, information on the choice of priors and Markov chain Monte Carlo settings |
| ☐ | ☒ | For hierarchical and complex designs, identification of the appropriate level for tests and full reporting of outcomes |
| ☒ | ☐ | Estimates of effect sizes (e.g. Cohen's *d*, Pearson's *r*), indicating how they were calculated |

*Our web collection on statistics for biologists contains articles on many of the points above.*

## Software and code

Policy information about availability of computer code

| Data collection | Illumina NovaSeq 6000(Illumina Inc., San Diego, CA, USA) |
|---|---|
| Data analysis | Software used: ADMIXTOOLS (version 3.0), ADMIXTURE (version 1.23), bcftools (version 1.15.1), Beagle (version 5.5), CLUMPP (version 1.1.2), DeepVariant (v1.6.0), Distruct (version 1.1), GEVA (version 1), GLnexus (version 1.3.1), IntroBlocker (version 2), iTOL (version 7), leeHom (version 1.2.17), mapDamage (version 2.0.8), Minimap2 (version 2.24), MUMmer (version 4.0.0), Novosort (version 3.06.05), pbmm2 (v1.10.0), Perl(V5.38.1), PLINK (version 1.9), PLINK2 (version 2.00a3.3LM), PopLDdecay (version 3.42), PSMC (version 0.6.5-r67), R (version 3.5.1), SAMtools (version 1.16.1), smartpca (version 7.2.1), SnpEff (version 4.3t). <br><br> Custom code availability: All custom scripts used in this study are available at https://github.com/guoyu-meng/barley-haplotype-script. |

For manuscripts utilizing custom algorithms or software that are central to the research but not yet described in published literature, software must be made available to editors and reviewers. We strongly encourage code deposition in a community repository (e.g. GitHub). See the Nature Portfolio guidelines for submitting code & software for further information.

## Data

Policy information about availability of data

All manuscripts must include a data availability statement. This statement should provide the following information, where applicable:

- Accession codes, unique identifiers, or web links for publicly available datasets
- A description of any restrictions on data availability
- For clinical datasets or third party data, please ensure that the statement adheres to our policy

> The sequence data collected in this study have been deposited at the European Nucleotide Archive (ENA)79 under BioProjects PRJEB65046, PRJEB56087 and PRJEB53924. The SNP and indel variant matrix are available at the European Variation Archive80 (EVA) under BioProject PRJEB79752. ENA accession codes for individual genotypes are listed in Supplementary Table 1. AHG matrices have been deposited in the Plant Genomics & Phenomics Research Data Repository81 under the DOI: http://doi.org/10.5447/ipk/2025/7.

## Research involving human participants, their data, or biological material

Policy information about studies with human participants or human data. See also policy information about sex, gender (identity/presentation), and sexual orientation and race, ethnicity and racism.

| | |
|---|---|
| Reporting on sex and gender | Not applicable. |
| Reporting on race, ethnicity, or other socially relevant groupings | Not applicable. |
| Population characteristics | Not applicable. |
| Recruitment | Not applicable. |
| Ethics oversight | Not applicable. |

Note that full information on the approval of the study protocol must also be provided in the manuscript.

# Field-specific reporting

Please select the one below that is the best fit for your research. If you are not sure, read the appropriate sections before making your selection.

☐ Life sciences  ☐ Behavioural & social sciences  ☒ Ecological, evolutionary & environmental sciences

For a reference copy of the document with all sections, see nature.com/documents/nr-reporting-summary-flat.pdf

# Ecological, evolutionary & environmental sciences study design

All studies must disclose on these points even when the disclosure is negative.

| | |
|---|---|
| Study description | The evolutionary history of barley domestication. |
| Research sample | 380 wild barley and 302 domesticated barley accessions. |
| Sampling strategy | Selection based on preliminary population structure |
| Data collection | Genebank accessions, DNA sequencing |
| Timing and spatial scale | Not applicable |
| Data exclusions | Admixed samples were excluded from downstream analysis as described in the manuscript. |
| Reproducibility | WGS data were compared with genebank genomics GBS data to confirm sample identities. |
| Randomization | Not applicable. |
| Blinding | Not applicable. |

Did the study involve field work?  ☐ Yes  ☒ No

# Reporting for specific materials, systems and methods

We require information from authors about some types of materials, experimental systems and methods used in many studies. Here, indicate whether each material, system or method listed is relevant to your study. If you are not sure if a list item applies to your research, read the appropriate section before selecting a response.

## Materials & experimental systems

| n/a | Involved in the study |
|-----|----------------------|
| ☒ | Antibodies |
| ☒ | Eukaryotic cell lines |
| ☐ ☒ | Palaeontology and archaeology |
| ☒ | Animals and other organisms |
| ☒ | Clinical data |
| ☒ | Dual use research of concern |
| ☐ ☒ | Plants |

## Methods

| n/a | Involved in the study |
|-----|----------------------|
| ☒ | ChIP-seq |
| ☒ | Flow cytometry |
| ☒ | MRI-based neuroimaging |

## Palaeontology and Archaeology

**Specimen provenance**
We analyzed ancient DNA sequences of 23 barley grains excavated at three archaeological sites in Israel (Supplementary Table 11). This number included published data of 5 barley grains from Yoram Cave. Archaeobotanical procedures were performed as described by LevMarom et al. Yoram Cave and Timna Valley Site have been described by Mascher et al. and Lev-Marom et al. Abi'or Cave is a medium-sized cave located on the eastern slopes of the Judean Desert, above Jericho, approximately 50 meters below sea level, across from the Karantal Monastery. The excavaEons at the cave were directed by the late H. Eshel in 1986. It is situated above a larger cave known as "The Spies Cave". The cave contains a main long tunnel, approximately 50 meters long, and has revealed archaeological material daEng from the Chalcolithic period to the Eme of the Bar Kokhba Revolt (2nd century CE). The cave was found to be heavily disturbed by animals, anEquiEes robbers, and monks who lived in it during the Islamic and more recent periods.

**Specimen deposition**
Yoram Cave, Timna Valley Site 34 and Abi'or Cave.

**Dating methods**
Radiocarbon dating of single seeds

☒ Tick this box to confirm that the raw and calibrated dates are available in the paper or in Supplementary Information.

**Ethics oversight**
Not applicable

Note that full information on the approval of the study protocol must also be provided in the manuscript.

## Plants

**Seed stocks**
Leibniz Institute of Plant Genetics and Crop Plant Research (IPK)

**Novel plant genotypes**
Not applicable

**Authentication**
WGS data were compared with genebank genomics GBS data to confirm sample identities.

