## [Peer Review file · Nature]

A haplotype-based evolutionary history of barley domestication

Corresponding Author: Dr Martin Mascher

Version 0:

Reviewer comments:

Referee #1

(Remarks to the Author)

This manuscript presents an extensive study on the evolutionary history of barley domestication based on a large-scale analysis of barley diversity and population structure. Using genomic sequences of a large set of wild, domesticated and ancient barley genotypes, the authors use a haplotype-based approach to identify spatiotemporal origins of haplotypes and to determine their contributions in early domesticated genotypes, in later gene flow and finally in the “modern” barley genotypes. The findings largely clarify the early events in domestication as well as later gene flow. The data provides strong evidence that haplotype diversification and adaptation occurred mostly independently, which is a finding of great importance for future work on the identification of adaptive loci. The authors conclude that the complex events in barley domestication and the later history of barley evolution complicate the identification of adaptive regions in the genome: a combination with mutational genomics will be necessary for their identification.

This is the first comprehensive analysis of barley domestication based on a large genomic dataset of more than 600 barley genotypes (wild, domesticated and, very importantly, ancient samples) and using a haplotype-based approach. Consequently, the findings described in the work are of high novelty and represent a leap in quality for the field of crop plant domestication. There are a large number of interesting findings (e.g. population bottlenecks before domestication), which are not repeated here. The complexity of events revealed by haplotype analysis clarifies many of the earlier, preliminary studies on barley domestication based on small datasets (single gene analysis, molecular markers, genome information of relatively few genotypes, or low sequence coverage of studied genotypes, and absence of haplotype-based analysis). Earlier studies indicated a mosaic genomic ancestry but did not provide a coherent explanation of the findings. The study here goes far beyond this earlier work by including haplotype analysis to study population structure and diversity.

Revisions

Can the authors comment on the limitations of absolute age determination of haplotypes, i.e. number of SNPs vs. age. Can the age determinations of ancient DNA/archaeological samples be used to (re-?) calibrate or confirm the mutation rate used as the basis for calculation of absolute age? I think the authors have an ideal dataset to explore this question.

There are some problems with the graphical description of the data and the presentation in the manuscript:

Figure 1a: the pie charts are hardly visible, unless zooming in massively. This is even more pronounced in Extended Data Figure 1b.

1b: the information superimposed on the map is hardly visible (population names)

Extended Data Figure 2a: “Pie charts to the right...” There are no pie charts in this figure.

The manuscript is very dense, and the authors should explore if the main findings of the paper can be presented more clearly (some comments on graphics are given above). Some extended data figures are not very useful. The type of data presented seems to fit better in a data repository than a journal article (e.g. some aspects of Extended Figure 5; Extended Figure 7). Similarly, Figure 3b. Summarizing this comment, I encourage the authors to focus on the main findings and

conclusions.

Figure 3e: the concept of frequency bins is not sufficiently explained. Does this figure show a chromosome?

Figure 5c: It is not clear which data were used for this graph.

L279 vs. Extended Data Figure 9a: The text states that the btr1 haplotype has an estimated age of 25 ka BP, the figure shows a peak at 10 ka BP. Please clarify.

Rephrase L116: We took advantage... Overly complex sentence. Formulate a research question.

L205: This pattern... What does "this" refer to?

Typos. Figure legend 1: ... 19788 domesticated barley GENOTYPES

L73: Should this be: reduced REpresentation sequencing... ?

L319: populationS

L358 : Rephrase sentence. "As current the... »

Referee #2

(Remarks to the Author)

Summary of the key results:

The authors used haplotype-based approaches to investigate the evolutionary history of wild and cultivated barley. They estimated ages of population size changes in wild barley and estimated the times when cultivated barley arose and from which wild populations. They examine their inferences in the context of data from archaeological samples and find concordance.

Originality and significance:

The analyses here help to refine what was already shown, but there doesn't seem to be any really novel or significant result or at least this is not clearly articulated.

Data and methodology, use of statistics:

The abstract claims that in this paper they sequenced the genomes of 623 genebank accessions and 23 archaeological specimens, but since some of these data were already published or are released as part of a co-submitted paper, I don't think they are correct. Based on the text in the Methods, it seems there are only 397 domesticated and 95 wild genebank accessions sequenced (which were sequenced to 3x coverage in a previous publication) and 18 new archaeological specimens.

Overall, the study was carefully conducted and well-documented in the Methods section, but I had concerns about some of the analyses/conclusions.

Major questions/concerns that I had were:

1) I was skeptical about the results for the proximal divergences in Figure 2a. Pericentromeric/centromeric regions tend to contain a high proportion of repeats, so that variant calls may be unreliable. Alternatively, if filtering is stringent, the regions may appear to be depauperate in variation. Either of these cases would likely lead to the type of noisy signal shown for the proximal regions.

This issue could also impact the patterns shown in Figure 3b and Extended Data Fig. 5b. Here, the conclusion is that there is lower diversity in the proximal regions relative to distal regions.

I would like to know more about the SNP density and missing data density (quality of calls) in the different regions. A recent paper published in Nature includes assemblies of 76 wild and domesticated barley accessions. (Some of the authors are the same.) The long-read data or assemblies could be used to validate some of the findings reported here. Also, showing the locations of centromeres should be shown since the patterns relative to the locations of the centromeres are mentioned in the text.

2) Why not use PSMC/MSMC to conduct the analysis of ancient population size? Instead, a window-based analysis was used. PSMC/MSMC breaks the genome into segments based on inferred recombination and then estimates the coalescence time of those segments. I didn't understand why the choice to use a simple window-based analysis was made, especially since PSMC was used in other contexts. I also did not understand why pseudo-diploids in the work leading to Figure 2b were produced by combining samples from different populations. Why not run PSMC on all pairs in each population as is done later for domesticated samples?

3) Conclusions about the long period of proto-domestication seem to hinge on allele age estimates, but it is unclear to me how the allele age estimates are inferred. Also, given the importance of these conclusions, it would be wise to use more robust approaches (or complementary approaches) to estimate allele ages.

Specific comments and suggested improvements:

Line 73: presentation -> representation

Lines 85-86: unclear how many accessions were newly sequenced here

Lines 93-95: Maybe a colonization bottleneck would contribute to this pattern?

Line 109: Why not conduct a more standard PSMC analysis? Why not separate by sub-population and plot the histories for those different populations together? What is gained from creating pseudo-diploids from different populations here?

Paragraph beginning with line 113: I am concerned that the different patterns observed for centromeric/pericentromeric regions could be due to the challenges for variant calling in these regions. Analyses of available long-read data might help to clarify what is going on here. (see point 1 above)

Lines 116-118: This sentence is confusing. Please revise.

Lines 122-124: I don't see why these two sentences are linked into a compound sentence; they are really two different thoughts.

Lines 136-139: I don't see any reason to invoke selection here, especially without any evidence, e.g., from population genetic simulations. It is often the case that old haplotypes persist.

Section beginning on line 146: Why not use PSMC rather than the simple window-based estimates? (see point 2 above)

Line 165: to make this point, it would be helpful if you labeled the centromeric regions on the plots (Fig. 3b and ED Fig. 5b).

Lines 176-177: I like the saturation analysis! It is nice to show that.

Section beginning at line 262:

An argument is made for a protracted proto-domestication period based on shared haplotypes with wild accessions as the divergence peaks are all between 0 and 10kya. But in the text, it is written the estimate for btr1 is 25kya, but in the figure it looks like the peak is just under 10kya, similar to the time of domestication. This would be the estimate based on mean pairwise differences, I believe. Given that the conclusions hinge on the allele age estimates, it would make sense to use complementary approaches, e.g., coalescent estimators in tsinfer or GEVA. Also, there is no information in the Methods about the allele age estimation procedure.

Conclusions:

Based on haplotype sharing, the authors suggest a protracted period of proto-domestication, but I do not see how that is concordant with the age estimates of the known domestication alleles. The question of whether the allele ages do or do not correspond to the splits of the domesticated lineages from the wild lineages is very interesting, but also requires a robust approach to estimating allele ages.

References:

As far as I can tell, the authors seem to give appropriate credit to previous work.

Clarity and context:

I thought the abstract, introduction and discussion could be revised to make the main points clearer.

Referee #3

(Remarks to the Author)

This is an impressive study which nicely builds on Poet's original observations of a mosaic domesticated genome with contributions from multiple wild barley ecotypes. I would like to see more of what each of these ecotypes provided in terms of adaptive robustness, but I suppose that is a follow-on study in itself.

I think this could be a suitable manuscript if substantial revisions were made, it is currently not suitable yet. The manuscript could do a better job of including the relevant literature in places. The archaeogenomics is not particularly effective here yet, and there are some nuances to analyses that could be better understood.

The general finding of an early onset of the selection process for barley in the 20-30K time window is exactly echoed in Allaby et al. 2017 (Phil. Trans. Roy. Soc. B 372: 20160429) based on genetic models of tracing selection strength and allele frequencies, and I believe was the first to use genetic based evidence to assert a very early onset of the processes leading to domestication. The same is true for einkorn and emmer too. Under the circumstances, this ought to be included in the discussion which currently simply refers to Snir's 2015 archaeobotanical evidence at Ohalo II and a study on rice.

Line 47-51: Many crop evolutionists operate, often tacitly, under the assumption that present-day population structure, especially in wild relatives, can reflect the situation thousands of years ago. No such proviso is needed for methods such as the pairwise sequentially Markovian coalescent (PSMC) that infer historic population size trajectories from genome sequences of extant individuals

The authors themselves go on to make this 'tacit' assumption in lines 82-83!

However, this statement is simply wrong and I worry whether the authors understand this analysis. It has been established for almost a decade now that PSMC suffers from artefacts consequent of population structure (Mazet et al 2016 *Heredity* 116, 362–371). Subdivided populations lead to an apparent declining N_e signal. This is in part due to the inflated heterozygosity caused by separate populations being treated as a single population (the basis of all F_{st} statistics, after all). Fortunately, the authors do not labor interpretations of N_e over time much in this manuscript, with the exception of the wild population. Remember that PSMC is a method that records the rate at which diversity is being generated over time (which obviously is a function of population size), not lost – however if one were to measure the N_e of the crop at that time, it would appear much higher because of the legacy diversity (if previous populations had been larger). The point here is that it

doesn't show a 'genetic bottleneck' – that statistic doesn't show loss of genetic diversity over time, but a very specific aspect of N_e . This sort of squares the hole of the observation that past crops have every time been shown to be very genetically diverse. However, it is likely that cultivated populations were highly subdivided into groups. This statistic is likely to equilibrate on the average size of group – ie it's not reflective of one population in the trough, but the size of the average population (of which there were many). If one examines Mazet's simulations (Figures 4 and 5), it is apparent that the N_e trough reflects deme size rather than total population size. So, the population sizes of around 20K (ish) that you see in Figure 6b, would equate to about a 5 hectare field systems going by sowing densities. The subsequent uptick associated with agricultural spread may be driven by population expansion and/or the increase in effective deme size through increase in communication – returning to something that is pretty close to the original wild population. This would seem to make a lot of sense in the archaeological context. Consequently, your data may give you more insights than you realized, but the statement on lines 47-51 has to go.

Line 58: Despite some successes^{10,11}, ancient plant DNA has had a comparatively lower impact on crop evolutionary studies owing to the poor preservation of plant material in most climates¹²

It is understandable that the authors cite themselves, but it is not credible to claim that they are responsible for the successes of plant archaeogenomics, or that there hasn't been substantial progress in the field – there are many higher profile studies than theirs that should have been cited here listed below. Furthermore, this is hardly an archaeogenome study anyway, a small relatively uninformative archaeogenome analysis is tacked on the end of the main study here.

Da Fonseca et al 2015 (Nat. Plants 1, 14003) - maize
Vallebuena-Estrada et al 2016 (Proc. Natl. Acad. Sci. U. S. A. 113, 14151–14156) - maize
Ramos-Madriral et al 2016 (Curr. Biol. 26, 3195–3201) - maize
Swarts et al 2017 (Science 357, 512–515) - maize
Kistler et al 2018 (Science 362, 1309–1313) - maize
Smith et al 2019 (Nat. Plants 5:369-379) – sorghum
Scott et al 2019 (Nat. Plants 5, 1120–1128) – wheat
Trucchi et al 2021 (Nat. Plants 7, 123–128) - beans
Ramos-Madriral et al 2025 (Cell 188, 1-11) - maize

Some further comments on the archaeogenomics. It is nice that some ancient genomes were included in the analysis, but they currently form a very minor component of this analysis. This is okay, but the analysis displayed in the main figure 5 is really clumsy, and looks to be a completely inappropriate analysis for what the authors appear to be trying to say. An $f_4 D$ statistic is applied here in what are patently inappropriate models. Forgive me, but I am used to the convention where P1 and P2 are the paired populations, P3 is the potential introgressing population and P4 is the outgroup – the authors appear to have gone for the reverse of this. In convention, a bias of P3 united with P2 gives a positive score (P2 to P4 in the manuscript). So, to take the Yoram Cave example, Israel barley forms a clade with the ancient sample from Israel more often than modern barleys from outside Israel do. The test is usually applied to identify disproportionate ancestry – what is this test achieving? It is not the least surprising that Ancient Israel barley is generally more closely related modern Israel barley than it is to Central Asian barley – P1 and P2 should not have been put together in this case (P3 and P4 in the manuscript) because they represent an incorrect model. If the authors wish to show that Yoram Cave barley is like Israel barley, then a phylogenetic tree would have been more appropriate – there is no apparent introgression question here. I would wager that if ISH-THS was in P1, Yoram P2, and everything else rotated through P3 (using the convention I am used to), you would get a zero score, because there is nothing here to see. This analysis appears meaningless to me, and frankly looks like the inclusion of an analysis that 'usually' occurs in these sorts of papers for the sake of it.

It would have made sense to see in more detail the state of haplotype content of the ancient genomes relative to the modern groups – a statement of how fixed the architecture was at that point, which I think is the general point the authors are trying to make.

Furthermore, diversity measures – (Supplementary Table 4) it is a pity that there are no measures of diversity presented for the ancient genomes that can be compared to the modern. While population diversity stats are presented for the modern pseudo population groupings which cannot be usefully applied to the single ancient genomes, other genome level statistics could be applied.

More minor points:

Extended figure 2: It is not immediately clear that panel 2b refers to distal, interstitial and proximal bins of genome (or indeed which is which), it is only implied in the main body of the text. I think the figure could be more clearly labelled here.

The ancient haplotype on chr 5H is intriguing. Have the authors scanned the gene content here or scanned for selection signatures (other than the general haplotype scan in Fig 3g -it's not clear what the window size is for this, if 5Mb then possibly not that sensitive)? I am not terribly surprised that the haploblocks themselves are not great at identifying selection signatures.

In dealing with the wild barley populations, I didn't see whether barleys representative of each pseudo population were filtered on admixture statistics to get a 'cleaner' signal. All groups shown in Figure 1a have members with partial mixed ancestry signals at $K=5$. This obviously would be expected to impact all analyses.

Line 212 – I suspect this should be Central Asia, not Western Asia. I'm not sure these values haven't been muddled – it looks from 4b that CA has more ancestral contribution than SD, whereas the text reports SD 16.3% and CA 12.9%. Other way round?

Version 1:

Reviewer comments:

Referee #1

(Remarks to the Author)

This is a revised version. While I checked mostly on the responses to my comments on the original submission, I also went through the changes made in response to the other two reviewers. The manuscript has very much improved by the revision. Data presentation and clarity of figures as well as more details on the analysis have improved the paper. Furthermore, the authors have added new types of analysis and replaced some parts of the work by methods identified as more appropriate by the reviewers.

The authors have addressed all my comments on the original submission. Several figures have been improved for clarity of data presentation and more information on the type of analysis has been provided. Moreover, the data of the full haplotype matrix have been deposited. This has improved the overall presentation of the work.

The analysis on the absolute age determination has been performed and I agree with the authors that their choice to use the mutation rate in *Brachypodium* is a very good proxy given that there is no information on mutation rate specifically for barley.

Referee #2

(Remarks to the Author)

The authors did a nice job of addressing my suggestions and concerns. I am satisfied with the revised manuscript and have no further suggestions.

Referee #3

(Remarks to the Author)

I appreciate the authors have seriously taken on board my comments, as well as those from the other reviewers, and I think this is a much improved manuscript as a consequence. This looks good to go now to me, and congratulations on an important and impressive study that helps push the frame of focus into deep time.

Reviewer #1

This manuscript presents an extensive study on the evolutionary history of barley domestication based on a large-scale analysis of barley diversity and population structure. Using genomic sequences of a large set of wild, domesticated and ancient barley genotypes, the authors use a haplotype-based approach to identify spatiotemporal origins of haplotypes and to determine their contributions in early domesticated genotypes, in later gene flow and finally in the “modern” barley genotypes. The findings largely clarify the early events in domestication as well as later gene flow. The data provides strong evidence that haplotype diversification and adaptation occurred mostly independently, which is a finding of great importance for future work on the identification of adaptive loci. The authors conclude that the complex events in barley domestication and the later history of barley evolution complicate the identification of adaptive regions in the genome: a combination with mutational genomics will be necessary for their identification.

This is the first comprehensive analysis of barley domestication based on a large genomic dataset of more than 600 barley genotypes (wild, domesticated and, very importantly, ancient samples) and using a haplotype-based approach. Consequently, the findings described in the work are of high novelty and represent a leap in quality for the field of crop plant domestication. There are a large number of interesting findings (e.g. population bottlenecks before domestication), which are not repeated here. The complexity of events revealed by haplotype analysis clarifies many of the earlier, preliminary studies on barley domestication based on small datasets (single gene analysis, molecular markers, genome information of relatively few genotypes, or low sequence coverage of studied genotypes, and absence of haplotype-based analysis). Earlier studies indicated a mosaic genomic ancestry but did not provide a coherent explanation of the findings. The study here goes far beyond this earlier work by including haplotype analysis to study population structure and diversity.

Revisions

Can the authors comment on the limitations of absolute age determination of haplotypes, i.e. number of SNPs vs. age. Can the age determinations of ancient DNA/archaeological samples be used to (re-?) calibrate or confirm the mutation rate used as the basis for calculation of absolute age? I think the authors have an ideal dataset to explore this question.

Answer: The limitations of absolute age determination of haplotypes include:

1. Short reads cannot be mapped uniquely to repetitive regions, resulting in the inability to call SNPs in these regions.
2. The spontaneous mutation rate of barley is not known.

To address issue 1, we employed a normalization-based approach. Specifically, we defined the effectively covered region between each pair of samples and performed normalization based on that region. To our knowledge, this is the most suitable solution to resolve the issue. Korunes and Samuk (2021, <https://doi.org/10.1111/1755-0998.13326>) employed a similar strategy to obtain unbiased estimates of nucleotide diversity and divergence. Their

calculations rely on all sites in the VCF file, including both polymorphic and monomorphic sites, which effectively represent uniquely mapped regions, as SNP calling is based on uniquely mapped reads.

To address issue 2, we adopted the spontaneous mutation rate from *Brachypodium distachyon*, a species closely related to barley, as previously published (Wang et al. 2019, PMID: 30964866). This study provides a genome-wide average natural mutation rate, which is particularly suitable for our genome-scale inference. Note that the spontaneous mutation rate of *B. distachyon* may still differ to some extent from the true mutation rate in barley.

We greatly appreciate your suggestion to use ancient DNA as a validation method. In response, we conducted a test using JK3014, an ancient barley sample dated to 6,000 years ago. Our phylogenetic analysis shows that JK3014 is most closely related to modern Israel two-rowed barley (ISR-SHS), and the two share a recent common ancestor, with their divergence likely occurring slightly before 6,000 years ago. Using the method outlined in our manuscript, we estimated the divergence time between JK3014 and ISR-SHS to be approximately 7,000 years, which aligns well with our expectations and further supports the accuracy of our approach (see **Supplementary Fig. 30**).

There are some problems with the graphical description of the data and the presentation in the manuscript:

Figure 1a: the pie charts are hardly visible, unless zooming in massively. This is even more pronounced in Extended Data Figure 1b.

Answer: We have revised the figure by using only the color of each sample's major ancestral component to plot the points. The points should now be clearly visible. For **Extended Data Figure 1b**, due to the wide longitudinal but narrow latitudinal distribution of our sampling locations, the map appeared overly compressed. To improve clarity, we have moved this figure to **Supplementary Figure 1** and provided a separate zoomed-in view of the Fertile Crescent region to enhance resolution.

1b: the information superimposed on the map is hardly visible (population names)

Answer: To improve clarity, we have increased the font size of the population labels and, wherever possible, moved them outside the corresponding color blocks.

Extended Data Figure 2a: "Pie charts to the right..." There are no pie charts in this figure.

Answer: We apologize that the pie charts were barely visible in the original version. To address this, we have now colored the samples named of the unadmixed individuals.

The manuscript is very dense, and the authors should explore if the main findings of the paper can be presented more clearly (some comments on graphics are given above). Some extended data figures are not very useful. The type of data presented seems to fit better in a data repository than a journal article (e.g. some aspects of Extended Figure 5; Extended Figure 7). Similarly, Figure 3b. Summarizing this comment, I encourage the authors to focus

on the main findings and conclusions.

Answer: We appreciate the reviewer's concern regarding the density of the manuscript and the complexity of some figures, including **Fig. 3b** and **Extended Data Figs. 5 and 7**. We acknowledge that these panels are visually complex, but they represent essential components of our study.

The construction of ancestral haplotype matrices and the genome-wide mapping of haplotype origins and ages are central to our analysis. Given the multidimensional nature of the underlying data, these figures are necessarily schematic — serving as guides to this complexity rather than exhaustive representations. To support transparency and reproducibility, we have deposited the full haplotype matrix in a public data repository (temporary DOI: <https://doi.ipk-gatersleben.de/DOI/a8515674-5c48-41a6-a662-1cb6b7cfff6c/f2a9c92f-972a-4256-be96-e5f028ecdc77/2/1847940088>).

We also note that the IntroBlocker approach is novel and, to our knowledge, has not been widely applied to crops. Consequently, the visual output may be less familiar to readers than more widely used population genomic methods (e.g., GWAS or selective sweep analyses), which further motivates the inclusion of explanatory visualizations.

Although we do not emphasize this aspect in the main text, the dataset underlying this study represents the most comprehensive whole-genome sequencing resource for wild and domesticated barley to date. We are currently working on adapting a genome browser to make these data more accessible and to complement the SNP-based variant matrices already available.

Figure 3e: the concept of frequency bins is not sufficiently explained. Does this figure show a chromosome?

Answer: The term “frequency bin” refers to intervals of haplotype frequency. Observed haplotype frequencies were grouped into 20 equal-width bins, each spanning 0.05 (i.e., 0–0.05, 0.05–0.1, ..., 0.95–1.0). The figure represents data from the entire genome, encompassing all seven barley chromosomes. This analysis is analogous to calculating the site frequency spectrum for SNPs.

We have expanded the figure legend accordingly.

Figure 5c: It is not clear which data were used for this graph.

Answer: This figure was drawn mainly based on the following data:

- (1) Liu et al. (2019, <https://doi.org/10.1016/j.quascirev.2018.12.017>). Their **Fig. 2** maps the dispersal route of domesticated barley from the Fertile Crescent to Central Asia, Europe, and Ethiopia before 7 ka BP, but without specific timing estimates.
- (2) For China, we supplemented routes to Tibet and East Asia based on Zeng et al. 2018 (<https://doi.org/10.1038/s41467-018-07920-5>).

(3) Taking into account the above information (1) and (2), along with our own divergence-time estimates (**Fig. 5a,b**) and population-structure results (**Supplementary Figs. 7,8**), we constructed this figure.

L279 vs. Extended Data Figure 9a: The text states that the *btr1* haplotype has an estimated age of 25 ka BP, the figure shows a peak at 10 ka BP. Please clarify.

Answer: Our method only used samples carrying the *btr1* allele, so the right boundary of the distribution (i.e., the sample pair with the earliest divergence) was interpreted as the emergence time of *btr1*. The same approach was applied to other gene mutations. Reviewer 3 suggested that we use published tools such as GEVA to estimate the timing. We tested GEVA and obtained the same result for *btr1* (~25 ka BP). Therefore, we removed our previous method and adopted GEVA instead.

Rephrase L116: We took advantage... Overly complex sentence. Formulate a research question.

Answer: We have revised this sentence:

“To better understand this pattern, we asked whether the divergence of long centromeric haplotypes reflects the divergence between individuals and the split times between populations.”

L205: This pattern... What does “this” refer to?

Answer: We have revised this sentence:

“The mosaic structure appears to have emerged early in the evolution of cultivated barley.”

Typos. Figure legend 1: ... 19788 domesticated barley GENOTYPES

L73: Should this be: reduced REpresentation sequencing... ?

L319: populationS

L358 : Rephrase sentence. “As current the... »

Answer: We have corrected these typos.

Reviewer #2

Summary of the key results:

The authors used haplotype-based approaches to investigate the evolutionary history of wild and cultivated barley. They estimated ages of population size changes in wild barley and estimated the times when cultivated barley arose and from which wild populations. They examine their inferences in the context of data from archaeological samples and find concordance.

Originality and significance:

The analyses here help to refine what was already shown, but there doesn't seem to be any

really novel or significant result or at least this is not clearly articulated.

Answer: We appreciate the reviewer's comment and have carefully revised the Abstract, Introduction, and Discussion to more clearly articulate the novelty and significance of our study. Specifically, we now emphasize how our haplotype-based approach builds on, but goes beyond, previous models of barley domestication. By integrating a large dataset of whole-genome sequences from both wild and domesticated accessions—including ancient DNA—we map the spatiotemporal origins of haplotypes across the barley genome. This allows us to resolve fine-scale ancestry patterns, clarify the timing and geography of early domestication events, and identify the persistence of gene flow long after domestication. We believe this genome-wide, haplotype-level resolution—applied at this scale and in a crop species—is a significant advance and clearly differentiates our work from earlier studies.

Data and methodology, use of statistics:

The abstract claims that in this paper they sequenced the genomes of 623 genebank accessions and 23 archaeological specimens, but since some of these data were already published or are released as part of a co-submitted paper, I don't think they are correct. Based on the text in the Methods, it seems there are only 397 domesticated and 95 wild genebank accessions sequenced (which were sequenced to 3x coverage in a previous publication) and 18 new archaeological specimens.

Answer: We have revised the sentence in the abstract to acknowledge the fact that we combine newly generated with previously published data:

We analysed the genomes of 682 genebank accessions and 23 archaeological specimens.

Additionally, in **Supplementary Table 1**, we added a column labeled 'Source' to indicate the origin of each sample (i.e., whether it was previously published or newly sequenced in this study).

Overall, the study was carefully conducted and well-documented in the Methods section, but I had concerns about some of the analyses/conclusions.

Major questions/concerns that I had were:

1) I was skeptical about the results for the proximal divergences in Figure 2a. Pericentromeric/centromeric regions tend to contain a high proportion of repeats, so that variant calls may be unreliable. Alternatively, if filtering is stringent, the regions may appear to be depauperate in variation. Either of these cases would likely lead to the type of noisy signal shown for the proximal regions.

This issue could also impact the patterns shown in Figure 3b and Extended Data Fig. 5b. Here, the conclusion is that there is lower diversity in the proximal regions relative to distal regions.

I would like to know more about the SNP density and missing data density (quality of calls) in the different regions. A recent paper published in Nature includes assemblies of 76 wild and domesticated barley accessions. (Some of the authors are the same.) The long-read data or

assemblies could be used to validate some of the findings reported here. Also, showing the locations of centromeres should be shown since the patterns relative to the locations of the centromeres are mentioned in the text.

Answer: The presence of long genomic segments with extremely low recombination rates in the centromeric regions is a hallmark feature of cereal genomes, such as barley, and is much less common in other species like humans. The phenomenon of reduced diversity in the proximal regions of cereal genomes has been reported in several previous studies, e.g. Mascher et al. (2017) <https://doi.org/10.1038/nature22043> in barley and Balfourier et al. (2019) <https://www.science.org/doi/10.1126/sciadv.aav0536> in wheat.

SNPs can only be called in regions covered by uniquely mapped reads, which we refer to as the effectively covered region. For large and complex genomes like barley, the greatest challenge in using short-read data to calculate SNP numbers between pairs of samples is the inability to achieve full coverage within a given window (e.g., 1 Mb). Therefore, we adopted a normalization strategy to calculate SNP counts (see **Supplementary Fig. 24**).

The proximal regions in barley tend to have larger effectively covered regions, making SNP number estimates more reliable in these regions. This is likely due to the fact that transposable elements (TEs) in these regions are older and more degraded, resulting in a higher proportion of unique sequences. In contrast, TEs in distal regions are younger (Mascher et al. 2017, PMID: 28447635), and the long-terminal repeat sequences of many inserted elements are highly similar, leading to a reduced proportion of unique sequences.

Thirteen of our samples overlap with the 76 published barley pan-genomes, for which high-fidelity (HiFi) long-read data are available. We used these 13 samples to compare SNP numbers obtained from HiFi reads versus short reads, with the following conclusions (see **Supplementary Fig. 25-27, Supplementary Table 20**):

(1) Calculating SNP density is not meaningful in this context, as pairwise SNP counts reflect differences between sample pairs, and a high SNP count does not necessarily indicate high diversity (e.g., if both carry the same alternative allele).

(2) Our SNP number calculation only involved filtering based on SNP call quality, without applying any population-level filtering. Later analysis of missing data revealed that the amount of missing information is low in both distal and proximal regions.

(3) Analysis of the effective covered region showed that coverage is actually highest in proximal regions. As a result, normalization requirements are minimal in these regions, and the SNP numbers are the most accurate.

(4) We compared the chromosomal distribution of SNPs based on short-read and HiFi-read data and found highly consistent patterns between the two.

In summary, the distribution of SNP numbers in proximal regions reported in our study is robust and reliable.

2) Why not use PSMC/MSMC to conduct the analysis of ancient population size? Instead, a

window-based analysis was used. PSMC/MSMC breaks the genome into segments based on inferred recombination and then estimates the coalescence time of those segments. I didn't understand why the choice to use a simple window-based analysis was made, especially since PSMC was used in other contexts. I also did not understand why pseudo-diploids in the work leading to Figure 2b were produced by combining samples from different populations. Why not run PSMC on all pairs in each population as is done later for domesticated samples?

Answer: The issue you raised appears to pertain to wild barley. The Pairwise Sequentially Markovian Coalescent (PSMC) method infers the changes in effective population size over time based on a single diploid genome. The model was originally developed for humans, assuming a relatively uniform recombination rate across the genome. However, in barley, recombination rates are extremely low in proximal (near-centromere) regions, and the evolutionary dynamics vary significantly across different chromosomal regions. To address this, we employed a window-based analysis to reveal the distinctive recombination landscape of barley chromosomes. Based on SNP distributions in the proximal regions, we observed that the divergence among wild barley populations is complex and likely occurred multiple times. In contrast, the population history inferred by PSMC—an average over the whole genome—fails to capture these finer-scale patterns, despite some methodological similarities between the two approaches.

Moreover, the PSMC model assumes random mating, whereby a single diploid individual carries diverse ancestral haplotypes. This assumption does not hold in selfing species like barley, which are nearly homozygous and thus lack the heterozygosity required for standard PSMC analysis. To overcome this limitation, researchers typically simulate a panmictic model by merging two individuals into a pseudo-diploid genome (see **references 31-33** from the **Online Methods**). This workaround highlights a common challenge in studying selfing species: many widely used tools were developed based on human genomic data and require adaptation for use in inbreeding organisms.

In our study, the PSMC results for wild barley were used in two main contexts: first, to help explain why the peak of SNP divergence in the distal regions among wild barley sample pairs is unimodal, which may be related to a bottleneck event during that period; second, as a comparative background for domesticated barley, in order to infer the timing of domestication. Our intention was to support our conclusions using multiple lines of evidence, and thus the PSMC analysis of wild barley was included as a supplementary line of support.

As per your suggestion, we performed two types of PSMC analyses. First, we conducted separate runs for each of the five wild barley subpopulations to infer their respective demographic histories. Second, we treated all wild barley samples as a single population to reconstruct the species-wide average demographic trajectory. However, it remains computationally infeasible to simulate every possible sample pair, as the number of combinations is exceedingly large ($C(251, 2) = 31375$). Given our current computational resources—and considering that such exhaustive sampling may not be necessary—we adopted a more efficient approach. We found that PSMC results were influenced by the pairwise identity-by-state (IBS) values between samples. PSMC assumes a panmictic (random mating) population in which the two haplotypes of a diploid individual originate from different ancestors. However, in selfing species like barley, this assumption is often

violated. When constructing pseudo-diploid genomes, sample pairs with high IBS values are more closely related and tend to share recent common ancestry, which undermines the random-mating assumption of the PSMC model. In contrast, pseudo-diploids constructed from pairs with lower IBS values are more likely to represent haplotypes from independent lineages, better satisfying the assumptions of the model and yielding more reliable demographic inferences. Therefore, we binned the sample pairs based on their IBS values and selected representative pairs from each bin for PSMC analysis. In total, we performed 773 PSMC runs.

For the population-specific PSMC analysis, we first calculated the IBS distribution for all pairwise combinations of individuals within each group. Based on the distribution, IBS values were divided into 2–4 bins. Within each bin, we selected either all sample pairs (if the number of combinations was fewer than 50) or 50 pairs (if the number of combinations was more than 50) evenly distributed from low to high IBS values. In selecting sample pairs, we also considered the sequencing coverage of each individual. A pair was retained only if the ratio of coverage, defined as $\text{ratio} = \text{coverage}_{\text{sample2}} / (\text{coverage}_{\text{sample1}} + \text{coverage}_{\text{sample2}})$, fell within the range 0.45–0.55. For the species-level PSMC analysis, the method was the same, except that each pair of samples was required to come from different subpopulations.

3) Conclusions about the long period of proto-domestication seem to hinge on allele age estimates, but it is unclear to me how the allele age estimates are inferred. Also, given the importance of these conclusions, it would be wise to use more robust approaches (or complementary approaches) to estimate allele ages.

Answer: We used the *btr1* allele as an example to illustrate our computational approach. Our method only used samples carrying the *btr1* allele, so the right boundary of the distribution (i.e., the sample pair with the earliest divergence) was interpreted as the emergence time of *btr1*. Admittedly, our previous approach was somewhat coarse and highly dependent on the choice of samples. Following your suggestion, we applied the published software GEVA and obtained the same estimate of ~25 ka BP for the *btr1* allele. Therefore, we decided to discard our original method and instead adopted the more robust GEVA approach.

Specific comments and suggested improvements:

Line 73: presentation -> representation

Answer: We have corrected this typo.

Lines 85-86: unclear how many accessions were newly sequenced here

Answer: We added a column in **Supplementary Table 1** labeled 'Source' to indicate the origin of each sample (i.e., whether it was previously published or newly sequenced in this study).

All 302 domesticated samples were newly sequenced. Eighteen of 23 analysed *ancient samples were newly sequenced*.

We added a sentence to the first paragraph of the Methods section to state how many wild barley samples were newly sequenced.

*Our wild barley panel (Supplementary Tables 1) is comprised of 285 accessions from the Wild Barley Diversity Collection (WBDC)^{1,2}, a collection of ecogeographically diverse accessions. The whole-genome sequencing of the WBDC collection is described in a companion paper³. A further 95 diverse barley accessions, mainly from the panel of Russell et al.⁴, were also included. **The latter set of samples had been sequenced to ~3× coverage by Jayakodi et al⁵. In the present study, we re-sequenced 32 of these samples to increase their coverage to ~10×.***

Lines 93-95: Maybe a colonization bottleneck would contribute to this pattern?

Answer: We concur with this interpretation. The relevant section from the manuscript is:

Notably, divergence times were multimodal, and the peaks in the distribution aligned with fluctuations in global surface temperature (Fig. 2d, Extended Data Fig. 3a). This pattern may be attributable to repeated episodes of colonization of new habitats, contraction and potential loss of populations, recolonization and secondary contact between populations. For example, the common ancestor of the Syrian Desert, Northern Mesopotamian and Central Asian populations split from the Northern Levantine lineage around 120 ka BP when a warm climate may have created new habitats. The Northern Mesopotamian and Central Asian populations split around 17 ka BP. This is consistent with the paleoclimatic modelling of Jakob et al., according to whom wild barley was absent from Central Asia as recently as 21 ka BP. The old age of the Southern Levantine population (i.e. its early divergence from populations elsewhere) is consistent with that region's supposed status as a glacial refugium.

Line 109: Why not conduct a more standard PSMC analysis? Why not separate by sub-population and plot the histories for those different populations together? What is gained from creating pseudo-diploids from different populations here?

Paragraph beginning with line 113: I am concerned that the different patterns observed for centromeric/pericentromeric regions could be due to the challenges for variant calling in these regions. Analyses of available long-read data might help to clarify what is going on here. (see point 1 above)

Answer: Please refer to our answers above.

Lines 116-118: This sentence is confusing. Please revise.

Answer: We have revised this sentence:

Lines 122-124: I don't see why these two sentences are linked into a compound sentence; they are really two different thoughts.

Answer: We have split this sentence into two.

Lines 136-139: I don't see any reason to invoke selection here, especially without any evidence, e.g., from population genetic simulations. It is often the case that old haplotypes persist.

Answer: We have run XP-CLR, a composite likelihood method for detecting selective sweeps, to find evidence for selection in the pericentromeric region of chromosome 5H. The results are reported in **Supplementary Fig. 4** of the revised manuscript.

Section beginning on line 146: Why not use PSMC rather than the simple window-based estimates? (see point 2 above)

Answer: Please see the answer above.

Line 165: to make this point, it would be helpful if you labeled the centromeric regions on the plots (Fig. 3b and ED Fig. 5b).

Answer: Yes, we have now included centromere icons in the figure.

Lines 176-177: I like the saturation analysis! It is nice to show that.

Answer: Thank you!

Section beginning at line 262:

An argument is made for a protracted proto-domestication period based on shared haplotypes with wild accessions as the divergence peaks are all between 0 and 10kya. But in the text, it is written the estimate for btr1 is 25kya, but in the figure it looks like the peak is just under 10kya, similar to the time of domestication. This would be the estimate based on mean pairwise differences, I believe. Given that the conclusions hinge on the allele age estimates, it would make sense to use complementary approaches, e.g., coalescent estimators in tsInfer or GEVA. Also, there is no information in the Methods about the allele age estimation procedure.

Answer: Please refer to our answer above. We are grateful to you for pointing us to GEVA.

Conclusions:

Based on haplotype sharing, the authors suggest a protracted period of proto-domestication, but I do not see how that is concordant with the age estimates of the known domestication alleles. The question of whether the allele ages do or do not correspond to the splits of the domesticated lineages from the wild lineages is very interesting, but also requires a robust approach to estimating allele ages.

Answer: We have used GEVA to provide more robust estimates of allele ages.

References:

As far as I can tell, the authors seem to give appropriate credit to previous work.

Clarity and context:

I thought the abstract, introduction and discussion could be revised to make the main points clearer.

Answer: We have thoroughly revised the abstract, introduction and discussion.

Referee #3 (Remarks to the Author):

This is an impressive study which nicely builds on Poet's original observations of a mosaic domesticated genome with contributions from multiple wild barley ecotypes. I would like to see more of what each of these ecotypes provided in terms of adaptive robustness, but I suppose that is a follow-on study in itself.

I think this could be a suitable manuscript if substantial revisions were made, it is currently not suitable yet. The manuscript could do a better job of including the relevant literature in places. The archaeogenomics is not particularly effective here yet, and there are some nuances to analyses that could be better understood.

The general finding of an early onset of the selection process for barley in the 20-30K time window is exactly echoed in Allaby et al. 2017 (Phil. Trans. Roy. Soc. B 372: 20160429) based on genetic models of tracing selection strength and allele frequencies, and I believe was the first to use genetic based evidence to assert a very early onset of the processes leading to domestication. The same is true for einkorn and emmer too. Under the circumstances, this ought to be included in the discussion which currently simply refers to Snir's 2015 archaeobotanical evidence at Ohalo II and a study on rice.

Answer: We thank the reviewer for pointing to the Allaby et al. study. We included it as reference #29 and refer to it in the interpretation of the GEVA analysis (haplotype age estimates):

Our estimated age of 27 ka BP for the btr1 haplotype (Extended Data Fig. 9d) predates the earliest archaeobotanical remains of domesticated barley by some 17,000 years, but it is closer to the ~22 ka BP estimate from an archaeobotanical modeling study. It is not impossible that non-shattering barleys (and the causal haplotypes) languished as rare variants in the wild before early cultivators selected them for propagation. The btr2 haplotype originated around 15 ka BP, which is very close to the ~12 ka BP estimate from Allaby et al.

Line 47-51: Many crop evolutionists operate, often tacitly, under the assumption that present-day population structure, especially in wild relatives, can reflect the situation thousands of years ago. No such proviso is needed for methods such as the pairwise sequentially Markovian coalescent (PMSC) that infer historic population size trajectories from genome sequences of extant individuals

The authors themselves go on to make this 'tacit' assumption in lines 82-83!

However, this statement is simply wrong and I worry whether the authors understand this analysis. It has been established for almost a decade now that PMSC suffers from artefacts consequent of population structure (Mazet et al 2016 Heredity 116, 362–371). Subdivided populations lead to an apparent declining Ne signal. This is in part due to the inflated heterozygosity caused by separate populations being treated as a single population (the basis of all Fst statistics, after all). Fortunately, the authors do not labor interpretations of Ne over time much in this manuscript, with the exception of the wild population. Remember

that PSMC is a method that records the rate at which diversity is being generated over time (which obviously is a function of population size), not lost – however if one were to measure the N_e of the crop at that time, it would appear much higher because of the legacy diversity (if previous populations had been larger). The point here is that it doesn't show a 'genetic bottleneck' – that statistic doesn't show loss of genetic diversity over time, but a very specific aspect of N_e . This sort of squares the hole of the observation that past crops have every time been shown to be very genetically diverse. However, it is likely that cultivated populations were highly subdivided into groups. This statistic is likely to equilibrate on the average size of group – ie it's not reflective of one population in the trough, but the size of the average population (of which there were many). If one examines Mazet's simulations (Figures 4 and 5), it is apparent that the N_e trough reflects deme size rather than total population size. So, the population sizes of around 20K (ish) that you see in Figure 6b, would equate to about a 5 hectare field systems going by sowing densities. The subsequent uptick associated with agricultural spread may be driven by population expansion and/or the increase in effective deme size through increase in communication – returning to something that is pretty close to the original wild population. This would seem to make a lot of sense in the archaeological context. Consequently, your data may give you more insights than you realized, but the statement on lines 47-51 has to go.

Answer: Thank you very much for your interpretation and suggestions regarding the PSMC results. However, we would like to emphasize that PSMC is based on a model developed for humans, which assumes random mating. In such a model, a single diploid genome carries haplotypes from different ancestors, enabling the reconstruction of population history.

In contrast, barley is a highly selfing species, and its genome is nearly homozygous, meaning that a single sample does not contain sufficient heterozygosity for reliable PSMC inference. To address this, it is common practice to combine two individuals to construct a pseudo-diploid genome, thereby mimicking the assumptions of random mating (see **references 31-33** from the **Online Methods**).

The increase in the y-axis values (effective population size) observed in our PSMC results is often related to the origin of the pseudo-diploid pairs. When the two samples come from already diverged populations, combining them artificially increases heterozygosity after their divergence, which inflates the recent effective population size. As shown in our previous **Fig. 6b**, the pseudo-diploids were constructed from individuals belonging to three already differentiated groups (Europe, Ethiopia, and Eastern Asia). Therefore, the observed recent increase in effective population size is an artifact.

To avoid such potential misinterpretations, we have removed **Fig. 6b** from the revised version. In our study, PSMC analyses were included not as the main evidence, but to support our findings through multiple lines of evidence. Specifically, PSMC results for wild and domesticated barley were used in two contexts: first, to help explain the unimodal distribution of SNP divergence in the distal chromosomal regions, which may reflect a historical bottleneck; and second, as a comparison between wild and domesticated barley to infer the timing of domestication.

We have revised the introduction thoroughly and removed the statement in ll. 47-51 of the original manuscript.

Line 58: Despite some successes^{10,11}, ancient plant DNA has had a comparatively lower impact on crop evolutionary studies owing to the poor preservation of plant material in most climates¹²

It is understandable that the authors cite themselves, but it is not credible to claim that they are responsible for the successes of plant archaeogenomics, or that there hasn't been substantial progress in the field – there are many higher profile studies than theirs that should have been cited here listed below. Furthermore, this is hardly an archaeogenome study anyway, a small relatively uninformative archaeogenome analysis is tacked on the end of the main study here.

Da Fonseca et al 2015 (Nat. Plants 1, 14003) - maize

Vallebuena-Estrada et al 2016 (Proc. Natl. Acad. Sci. U. S. A. 113, 14151–14156) - maize

Ramos-Madrugal et al 2016 (Curr. Biol. 26, 3195–3201) - maize

Swarts et al 2017 (Science 357, 512–515) - maize

Kistler et al 2018 (Science 362, 1309–1313) - maize

Smith et al 2019 (Nat. Plants 5:369-379) – sorghum

Scott et al 2019 (Nat. Plants 5, 1120–1128) – wheat

Trucchi et al 2021 (Nat. Plants 7, 123–128) - beans

Ramos-Madrugal et al 2025 (Cell 188, 1-11) - maize

Answer: We appreciate the reviewer's comments and agree that our previous phrasing may have overstated the novelty and impact of our own contributions. In the revised manuscript, we have adopted a more balanced tone and now state: "Lastly, ancient DNA sequences provide valuable insights into past genetic diversity, although their use is limited by the often poor preservation of plant material in many climates." We have also cited the recent review by Kistler et al. (2020) PMID: 32119793, which provides a comprehensive overview of developments in ancient plant genomics. Our aim here is to acknowledge both the promise and the current limitations of ancient DNA in crop evolutionary studies, while situating our own work as a modest contribution to this growing field.

Some further comments on the archaeogenomics. It is nice that some ancient genomes were included in the analysis, but they currently form a very minor component of this analysis. This is okay, but the analysis displayed in the main figure 5 is really clumsy, and looks to be a completely inappropriate analysis for what the authors appear to be trying to say. An f4 D statistic is applied here in what are patently inappropriate models. Forgive me, but I am used to the convention where P1 and P2 are the paired populations, P3 is the potential introgressing population and P4 is the outgroup – the authors appear to have gone for the reverse of this. In convention, a bias of P3 united with P2 gives a positive score (P2 to P4 in the manuscript). So, to take the Yoram Cave example, Israel barley forms a clade with the ancient sample from Israel more often than modern barleys from outside Israel do. The test is usually applied to identify disproportionate ancestry – what is this test achieving? It is not the least surprising that Ancient Israel barley is generally more closely related modern Israel barley than it is to Central Asian barley – P1 and P2 should not have been put together in this case (P3 and P4 in the manuscript) because they represent an incorrect model. If the authors wish to show that Yoram Cave barley is like Israel barley, then a phylogenetic tree would have been more appropriate – there is no apparent introgression question here. I would

wager that if ISR-THS was in P1, Yoram P2, and everything else rotated through P3 (using the convention I am used to), you would get a zero score, because there is nothing here to see. This analysis appears meaningless to me, and frankly looks like the inclusion of an analysis that ‘usually’ occurs in these sorts of papers for the sake of it.

It would have made sense to see in more detail the state of haplotype content of the ancient genomes relative to the modern groups – a statement of how fixed the architecture was at that point, which I think is the general point the authors are trying to make.

Furthermore, diversity measures – (Supplementary Table 4) it is a pity that there are no measures of diversity presented for the ancient genomes that can be compared to the modern. While population diversity stats are presented for the modern pseudo population groupings which cannot be usefully applied to the single ancient genomes, other genome level statistics could be applied.

Answer: We thank the reviewer for their careful reading and constructive comments on our archaeogenomic analysis. We especially appreciate the critique regarding the application of D-statistics and the suggestion to more appropriately define “paired populations.” We have revised the analysis accordingly and made substantial improvements to both our methodological approach and the clarity of our interpretations.

1. Revision of D-statistics framework

We acknowledge the misuse of the D-statistic model in the original manuscript, particularly the inappropriate designation of P1 and P2. Based on the reviewer’s recommendation, we restructured the analysis by first defining the “paired populations” using multiple lines of evidence, including identity-by-state (IBS) clustering with GBS data and phylogenetic relationships. For example, Yoram Cave and Timna 34 (both two-rowed) cluster closely with ISR-THS, while Abi’or Cave shows genetic affinity to ME-SHS. These paired groupings were used consistently in the revised D-statistics framework.

We now follow the conventional D-statistic form: $D(P1, P2; P3, P4)$, where P1 and P2 are the paired populations and P4 is the outgroup (*H. pubiflorum*). The revised tests are as follows:

- $D(\text{ISR-THS, Yoram Cave}; P3, H. \text{ pubiflorum})$
- $D(\text{ISR-THS, Timna 34}; P3, H. \text{ pubiflorum})$
- $D(\text{ME-SHS, Abi’or Cave}; P3, H. \text{ pubiflorum})$

The results reveal no detectable gene flow between Yoram Cave and any P3 populations, consistent with its genetic closeness to ISR-THS and suggesting limited introgression. In contrast, Timna 34 shows gene flow with several western barley populations, while Abi’or Cave shows the strongest evidence of gene flow, particularly with populations from Europe and Ethiopia. This pattern likely contributes to its higher genetic diversity relative to the other ancient samples.

2. Haplotype and genomic structure of ancient barley

To better contextualize the ancient genomes, we carried out an analysis of haplotypes at domestication loci (*Btr* and *Vrs1*) and in pericentromeric regions. In most cases, the ancient samples carried haplotypes also found in modern domesticated barley, suggesting that much of the domesticated genomic architecture was already present. However, we also identified private haplotypes in the 7H pericentromeric region of some ancient barleys, pointing to localized variation.

Additionally, PCA and phylogenetic analyses suggest that ancient barley in Israel was not simply a direct descendant of local wild barley, but rather derived from a founder population, with subsequent regional differentiation and admixture.

3. Diversity metrics for ancient genomes

As the reviewer rightly noted, traditional population-based diversity measures are not well-suited to single ancient genomes. To address this, we developed an approach based on rare alleles ($MAF \leq 0.01$) from a large wild barley reference panel. These rare alleles serve as proxies for ancestral diversity. We then calculated the relative change in diversity between ancient and modern individual genomes using the formula: $(M - A)/A$, where M and A are the counts of rare alleles in the modern and ancient samples, respectively.

This method allows us to compare diversity at the individual level. The results show a gradient in diversity among the ancient samples: Yoram Cave < ISR-THS \approx Timna 34 < Abi'or Cave. This variation correlates well with the inferred levels of gene flow from the revised D-statistics.

4. Summary

Together, these revisions address the reviewer's concerns by:

- Correcting the D-statistics model and using appropriate paired populations.
- Replacing potentially misleading analyses with phylogenetic and haplotype-based methods.
- Implementing a novel method for comparing diversity in single ancient genomes.
- Providing stronger support for the narrative that gene flow—rather than domestication bottlenecks—played a significant role in shaping the observed diversity patterns in ancient barley.

We are grateful for the reviewer's guidance, which significantly improved the scientific rigor and interpretability of our archaeogenomic analysis.

More minor points:

Extended figure 2: It is not immediately clear that panel 2b refers to distal, interstitial and proximal bins of genome (or indeed which is which), it is only implied in the main body of the text. I think the figure could be more clearly labelled here.

Answer: We have indicated the specific boundaries of the three regions—distal, interstitial, and proximal—in **Extended Data Fig. 2b**.

The ancient haplotype on chr 5H is intriguing. Have the authors scanned the gene content here or scanned for selection signatures (other than the general haplotype scan in Fig 3g -it's not clear what the window size is for this, if 5Mb then possibly not that sensitive)? I am not terribly surprised that the haploblocks themselves are not great at identifying selection signatures.

Answer: We added a selective sweep analysis (XP-CLR) by comparing the Southern Levant (SL) population with the other four wild barley groups (**Supplementary Fig. 4**). Combined with the F_{st} results in **Extended Data Fig. 1d**, we confirmed that this region shows a clear signature of selection. We listed the potential candidate genes within this region (**Supplementary Table 5**); however, due to the large size of the region (approximately 90 Mb), it remains unclear which specific gene(s) were targeted by selection.

In dealing with the wild barley populations, I didn't see whether barleys representative of each pseudo population were filtered on admixture statistics to get a 'cleaner' signal. All groups shown in Figure 1a have members with partial mixed ancestry signals at K=5. This obviously would be expected to impact all analyses.

Answer: In our analyses of the five wild barley populations, we only included unadmixed individuals, defined as those whose primary ancestral component accounted for ≥ 0.85 . Please refer to **Supplementary Table 3** for the detailed definition and list of unadmixed samples.

Line 212 – I suspect this should be Central Asia, not Western Asia. I'm not sure these values haven't been muddled – it looks from 4b that CA has more ancestral contribution than SD, whereas the text reports SD 16.3% and CA 12.9%. Other way round?

Answer: Yes, the term "Western Asia" should indeed be corrected to "Northern Mesopotamia". Additionally, to improve clarity, we have directly added the specific percentages on **Fig. 4b**.